



# Optimized Stochastic Representation of Soil States Model Uncertainty of WRF (v4.2) in the Ensemble Data Assimilation System

Sujeong Lim[1,2], Seon Ki Park[1,2,3], and Claudio Cassardo[4]

[1]Center for Climate/Environment Change Prediction Research, Ewha Womans University, Seoul, 03760, Republic of Korea
[2]Severe Storm Research Center, Ewha Womans University, Seoul, 03760, Republic of Korea
[3]Department of Climate and Energy System Engineering, Ewha Womans University, Seoul, 03760, Republic of Korea
[4]Department of Physics and NatRisk Centre, University of Turin, Turin, 10125, Italy

**Correspondence:** Seon Ki Park (spark@ewha.ac.kr)

**Abstract.** The ensemble data assimilation (EDA) system represents the model uncertainties by ensemble spread that is a standard deviation of ensemble background error covariance (BEC). However, this ensemble spread is usually underestimated due to insufficient ensemble size, sampling errors, and imperfect models: it often causes a filter divergence problem as the analysis ignores the observation due to insufficient model uncertainty. This phenomenon is also found in the coupled land-atmospheric

modeling system, especially near the surface where the heat flux exchanges are crucial as the lower boundary conditions. We have developed the stochastic perturbations to soil states scheme (SPSS) within the coupled Weather Research and Forecasting-Noah Land Surface Model (WRF-Noah LSM) to represent the near surface uncertainty. It perturbs soil temperature and soil moisture, respectively, by adding the random forcing to inflate the ensemble spread. The random forcing used in perturbation is controlled by the tuning parameters such as amplitude, decorrelation length and time scale, which vary depending on the

target variables. To obtain the optimal random forcing tuning parameters to the soil states, we have implemented a global optimization algorithm — the micro-genetic algorithm, which operates the principles of natural selection or survival of the fittest to evolve the best potential solution having small populations. Optimizations are performed on daytime and nighttime to account for diurnal variations of soil states and evaluated by a fitness function that describes the interaction between the land and atmospheric systems in terms of accuracy. As a result, the soil temperature and soil moisture perturbations with SPSS

can indirectly inflate the ensemble BECs of temperature and water vapor mixing ratio in the planetary boundary layer (PBL) of the EDA system. The SPSS with diurnal variations depicts reasonable ensemble spread for soil states, but the ensemble spread for atmospheric states from the propagation of the soil states perturbations is less effective. Furthermore, the inflated soil temperature helps to produce an adequate analysis increment reducing the background error of temperature in PBL. Soil moisture, however, requires more prescriptions to generate an adequate analysis increment reducing the background error of

water vapor mixing ratio in PBL.


# 1 Introduction

The ensemble data assimilation (EDA) system describes both initial conditions (ICs) and model uncertainties represented by the flow-dependent background error covariance (BEC). The ensemble BEC is expressed by an ensemble spread ($\sigma(x)$), which is the standard deviation with respect to the ensemble mean ($\bar{x}$) and expected to be similar similar to the ensemble mean error

($e(\bar{x})$) (Fig 1a). The EDA in the numerical weather prediction (NWP) system generally suffers from the underestimated BEC due to limited ensemble size, sampling errors, and imperfect models (Miyoshi, 2011; Lim et al., 2020). As a result, it generates a filter divergence problem: the analysis diverges from the true state by ignoring the observation due to over-confidence in the forecast (Fig 1b). On the contrary, the overestimated BEC excessively relies on observation (Fig 1c).

The sufficient ensemble sizes can moderately remove the sampling error (Miyoshi, 2011; Kunii, 2014); however, it requires
enormous computational resources. The most common remedies for BEC deficiencies are covariance localization or inflation (Anderson, 2012; Anderson and Anderson, 1999; Mitchell and Houtekamer, 2000; Houtekamer and Mitchell, 2005; Zhang et al., 2004; Whitaker and Hamill, 2012; Ying and Zhang, 2015; Lim et al., 2020). The former does not use observations out of a cutoff distance from a state variable, and the latter inflates the covariance for analysis or background (i.e., forecast). In this study, we focus on the covariance inflation approach, especially accounting for the *model uncertainty* that arises during
integration (Ollinaho et al., 2017).

As one of the inflation methods, the stochastic perturbation scheme represents the model uncertainty by perturbing the tendencies or parameters in the physical parameterization schemes (Buizza et al., 1999; Shutts, 2005; Palmer et al., 2009): it assumes that the NWP model contains inevitable uncertainties in the physical processes by simplifications and approximations. For example, the stochastically perturbed parameterization tendency scheme (SPPT) assumes uncertainties in the parameter-
ized physical tendencies (Palmer et al., 2009); the stochastically perturbed dynamical tendencies scheme (SPDT) assumes uncertainties in the dynamical tendencies from computational representations of underlying partial differential equations (Koo and Hong, 2014). The stochastic kinetic energy backscatter scheme (SKEB) represents the model uncertainties for scale interactions that are absent in a truncated NWP model by perturbing the stream function and potential temperature tendencies (Shutts, 2005); the stochastically perturbed parameterizations scheme (SPP) represents uncertainties in the pararameteriza-
tion of physical processes (e.g., turbulent diffusion and subgrid orography, convection, clouds and large-scale precipitation, and radiation) via parameter perturbations. If these stochastic perturbation schemes are applied in the EDA system (e.g., Lim et al., 2020), they provide different realizations (e.g., the physics and dynamics tendencies) within each ensemble member by interacting with observational reality (Shutts et al., 2011).

This stochastic representations of model uncertainty can address a coupled process (e.g., atmosphere-land surface) where
a lack of spread exists in the near-surface variables (Leutbecher et al., 2017). In particular, the land surface model (LSM) interacts with the lower atmosphere as boundary conditions. Because it is strongly coupled to the atmospheric state at certain times and in certain places, the less land-surface uncertainty may lead to improvements in atmospheric forecasts (MacLeod et al., 2016). Several studies examined the impact on atmospheric ensembles by perturbing land parameters or soil states or the tendency of soil states (Orth et al., 2016; MacLeod et al., 2016; Draper, 2021). For example, the land parameters include

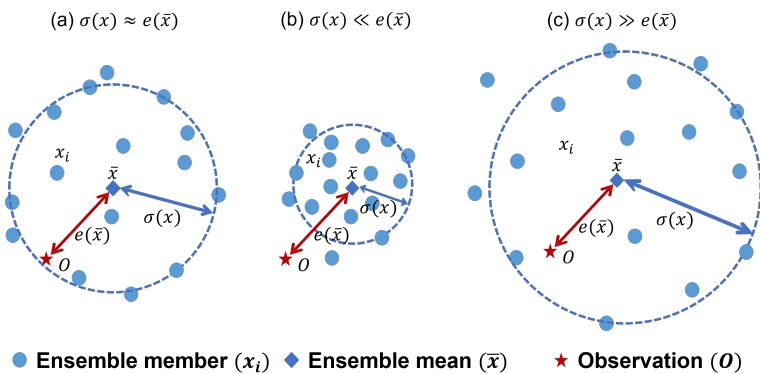

**Figure 1.** Schematic diagram of ensemble spread ($\sigma(x)$, blue arrow) and ensemble mean error ($e(\bar{x})$, red arrow) using the ensemble members ($x_i$; $i$ is the number of ensembles, blue circles), ensemble mean ($\bar{x}$, blue diamond), and observation ($O$, red star): (a) optimal ensemble spread, (b) under-disperse ensemble spread, and (c) over-disperse ensemble spread.

the minimum stomatal resistance, total soil depth, hydraulic conductivity, vegetation fraction, etc.; the soil states include soil temperature and soil moisture. In particular, the soil states directly affect the near-surface temperature and humidity forecasts through the sensible and latent heat fluxes response (Kim and Hong, 2007; Xue et al., 2012; Deng et al., 2016; Lin and Pu, 2020).

In this study, we hypothesize that the near-surface uncertainty can be represented by introducing stochastic perturbations
into the soil states in the Noah Land Surface Model (Noah LSM) coupled to the Weather Research and Forecast (WRF) model. Note that the land and atmosphere have different behaviors in dynamics and error growth (Draper, 2021); furthermore, the random forcing tuning parameters that determine the perturbations have not been completely understood (Lupo et al., 2020). To suggest the optimal random forcing tuning parameters used to perturb the soil states, we bring the optimization algorithm that automatically searches the best parameters in both soil and atmospheric states. We also investigate whether the inflation
of soil states can alter the near-surface atmospheric states in the EDA system considering the diurnal variations of soil states. Results are focused on the planetary boundary layer (PBL), which is the bottom part of the atmosphere directly influenced by the land surface and responds to surface forcing with a time scale of about an hour or less. Section 2 describes the methodology, section 3 introduces the data and experimental design and section 4 provides the results. Conclusions are summarized in section 5.

## 2   Methodology

### 2.1   WRF-Noah LSM Coupled System

This study used version 4.2 of the Advanced Research WRF (ARW) solver (Skamarock et al., 2019) as a mesoscale weather prediction system. Figure 2 shows the experimental WRF processing system (WPS) domain with the Lambert conformal

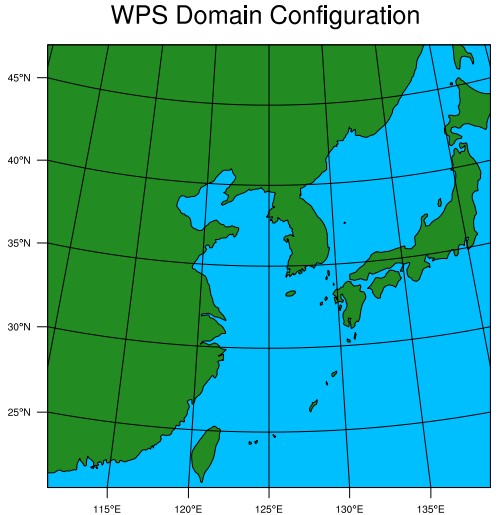

**Figure 2.** WPS domain configuration.

projection. The horizontal resolution was 30 km with $100 \times 100$ grid points. To investigate East Asia centered on the Korean

Peninsula, we simulated a single large domain and did not consider the perturbations of the lateral boundary conditions. The

model top sets to 50 hPa with 32 atmospheric and 4 soil layers. We employed the physics suite "TROPICAL", well tested for

tropical storms and tropical convection: it includes the WRF Single-Moment six-class (WSM6) microphysics scheme (Hong

and Lim, 2006), new Tiedtke cumulus parameterization scheme (Zhang and Wang, 2017), Rapid Radiative Transfer Model

for GCMs (RRTMG) shortwave and longwave radiation scheme (Iacono et al., 2008), YonSei University (YSU) planetary

boundary layer scheme (Hong et al., 2006), old MM5 surface layer scheme (Zhang and Anthes, 1982), and Noah Land Surface

Model (Chen and Dudhia, 2001; Ek et al., 2003).

In particular, Noah LSM constitutes a land component as the bottom condition of the atmospheric model (e.g., WRF), which

is associated with surface energy balance, soil heat flow, and soil water flow (Chen and Dudhia, 2001; Greve et al., 2013; Zheng

et al., 2015a, b). It calculates soil temperature, soil moisture, surface skin temperature, canopy water content, snowpack water

equivalent content, snowpack depth, and so on in the 4 soil layers — 10, 30, 60, and 100 cm thickness up to 2 m soil depth

(Chen et al., 1997; Ek et al., 2003). Among the prognostic variables, soil temperature ($T$; in K) is calculated by the thermal

diffusion equation:

$$C(\theta)\frac{\partial T}{\partial t} = \frac{\partial}{\partial z}[K_t(\theta)\frac{\partial T}{\partial z}] \tag{1}$$

where volumetric heat capacity ($C$; in J m$^{-3}$ K$^{-1}$) and thermal conductivity ($K_t$; in W m$^{-1}$ K$^{-1}$) are formulated as nonlinear

functions of volumetric soil moisture content ($\theta$; in m$^3$ m$^{-3}$), soil texture (Cosby et al., 1984) and soil depth ($z$; in m) (Chen

et al., 1996; Greve et al., 2013). The prognostic equation for $\theta$ is defined as the volume of soil water relative to the total volume





including soil, water, and air from the diffusive form of Richard's equation (Chen et al., 1996; Greve et al., 2013; Zheng et al., 2015a):

$$\frac{\partial \theta}{\partial t} = \frac{\partial}{\partial z}(D\frac{\partial \theta}{\partial z}) + \frac{\partial K}{\partial z} + F(\theta) \tag{2}$$

for each soil layer. Both soil water diffusivity ($D$; in m$^2$ s$^{-1}$) and hydraulic conductivity ($K$; in m s$^{-1}$) are a nonlinear function of $\theta$ and soil texture (Cosby et al., 1984), and $F_\theta$ is the sources and sinks (i.e., precipitation, evaporation, and runoff; in m s$^{-1}$).

The surface energy balance equation can be written as follows (see Hou et al., 2015; Huang et al., 2016):

$$R_{net} = H + LE + G_0, \tag{3}$$

where $R_{net}$ is net radiation, $H$ sensible heat flux, $LE$ latent heat flux, and $G_0$ ground heat flux (all in W m$^{-2}$). In the following,
we describe $H$, $LE$, and $G_0$ in more detail. First, $H$ is formulated as

$$H = \rho c_p C_h |u_{air}|(T_{sfc} - T_{air}) \tag{4}$$

where $\rho$ is density of air (in kg m$^{-3}$), $c_p$ is the specific heat capacity of air (in J kg$^{-1}$ K$^{-1}$), $C_h$ is the surface exchange coefficient for heat (unitless), $u_{air}$ is wind speed (in m s$^{-1}$), $T_{sfc}$ is potential temperature at the surface (in K), and $T_{air}$ is potential atmospheric temperature (in K) at the lowest model level (Chen et al., 1997; Hou et al., 2015; Zheng et al., 2015b; 
Huang et al., 2016). Second, $LE$ is represented as follows:

$$LE = \rho C_e |u_{air}|(q_{sfc} - q_{air}) \tag{5}$$

where $C_e$ is the surface exchange coefficient for water vapor (unitless), $q_{sfc}$ is surface specific humidity (in kg kg$^{-1}$), and $q_{air}$ is air specific humidity at the lowest model level (in kg kg$^{-1}$) (Chen et al., 1997). Lastly, $G_0$ is calculated through Fourier's law using the temperature gradient between the surface and the midpoint of the topsoil layer (Huang et al., 2016):

$$G_0 = K_t(\theta)(\frac{T_{sfc} - T_{s,1}}{\Delta z_1}) \tag{6}$$

where $T_{s,1}$ is temperature at the midpoint of the topsoil layer (in K).

## 2.2  Stochastic Perturbations to Soil States scheme (SPSS) in WRF-Noah LSM

We developed the Stochastic Perturbations to Soil States scheme (SPSS) for Noah LSM in WRF to resolve the near-surface uncertainties. The SPSS randomly perturbed soil temperature or soil moisture in the topsoil layer using a spatially and temporally
correlated random forcing (RF) at each grid point and time step as follows:

$$x_{i,new}^1 = x_i^1 + r_i \tag{7}$$



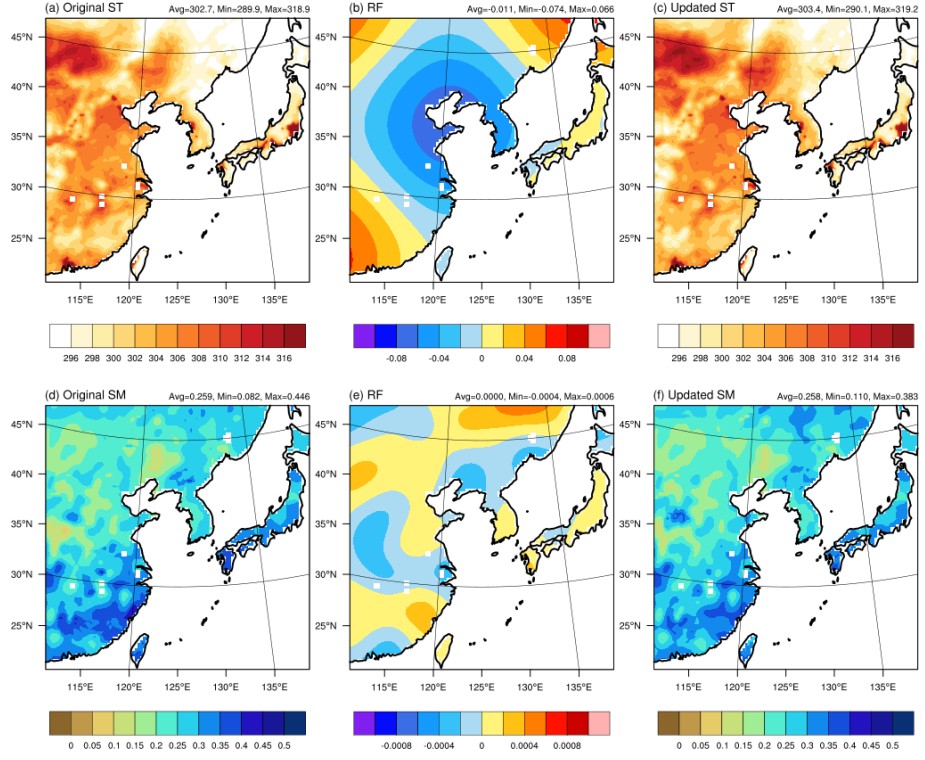

**Figure 3.** The 6 hour forecast fields of ensemble member 1 showing the effect of RF on soil temperature (ST in K; upper panels) and soil moisture (SM in $m^3$ $m^{-3}$; lower panels): (a) Original ST, (b) RF for ST (with $\sigma$ = 0.13 K, $L$ = 2900 km, and $\tau$ = 120 s), and (c) updated ST (i.e., original + RF); (d) original SM, (b) RF for SM (with $\sigma$ = 0.0003 $m^3$ $m^{-3}$, $L$ = 250 km, and $\tau$ = 900 s), and (f) updated SM.

where $x_i^1$ is soil temperature or soil moisture at the first soil layer (i.e., topsoil layer), $r$ is the RF following the Gaussian distribution with zero mean, and $i$ is the index for ensemble member. Note that we only perturbed the topsoil layer because the perturbations in the bottom layers generated worse performance, and the topsoil layer can sufficiently affect the ensemble

BEC as represented by the 6 hour forecast. The soil moisture perturbation requires special cautions, as introduced in Tennant and Beare (2014): 1) soil moisture with RF must be between the wilting point and saturation levels, and 2) the perturbations under snow cover or frozen soil are set to zero. Tennant and Beare (2014) addressed the IC perturbations, but these cautions are identically necessary for perturbing soil moisture in a given forecast time step.

The RF parameters used for perturbation are regarded as tuning parameters that depend on the characteristics being applied.

Each parameter is defined as follows (Lim et al., 2020): the amplitude ($\sigma$) determines the perturbation size of RF as a standard deviation of RF; the horizontal decorrelation length scale ($L$) determines how errors propagate in horizontal direction; the decorrelation time scale ($\tau$) determines how long the perturbed errors will be sustained. Figure 3 shows how RF changes soil temperature (Fig 3a-c) and soil moisture (Fig 3d-f), respectively.





As pointed out by Jin and Mullens (2014), our results also indicated stronger diurnal variations in soil temperature, being
colder at nighttime and warmer at daytime, than in soil moisture. Therefore, we investigated the effectiveness of diurnal
variation in RF tuning parameters to perturb soil temperature and soil moisture.

## 2.3    Optimization Tool: Micro-Genetic Algorithm ($\mu$-GA)

### 2.3.1    $\mu$-GA

The genetic algorithm (GA) is a global optimization algorithm developed by John Holland in the 1970s (e.g., Holland,
1973, 1975) and is based on Darwinian principles of natural selection (Golberg, 1989). It uses selection, crossover, and mu-
tation to operate a set of potential solutions called populations, expressed by the string called a chromosome or an individual.
Here, the chromosome is represented by the binary form called a gene. First, the selection operator chooses the good solution
or eliminates the bad solution based on the fitness value. Second, the crossover operator exchanges the genetic information
between solutions using single-point or uniform types. Lastly, the mutation operator modifies the value of each gene of the
chromosomes by replacing it with the opposite value, e.g., from 0 to 1: it helps to prevent premature convergence. When these
processes create a new generation, the above process is repeated until the convergence condition or the prescribed number of
generations (i.e., iterations) are satisfied.

The micro-genetic algorithm ($\mu$-GA) is a small population GA with re-initialization (Krishnakumar, 1990), whereas standard
GA uses large populations to secure diversity upon convergence (Yu et al., 2013); thus, $\mu$-GA requires less computational time
than standard GA. Moreover, $\mu$-GA does not use mutation to achieve diversity. Instead, it starts with a new individual by
re-initialization or uses elitism, which assigns the best individual among the small individuals based on the fitness evaluation
and carries it to the next generation; thus, it guarantees to preserve the good solutions during the generations. A previous study
showed that the small population could sufficiently reach the entire search space by crossover alone. The $\mu$-GA has been widely
used for optimal parameter estimation (e.g., Lee et al., 2006; Yu et al., 2013; Lim et al., 2022) or scheme-based optimization
(e.g., Hong et al., 2014, 2015; Park and Park, 2021; Yoon et al., 2021) in meteorology and hydrology.

### 2.3.2    Coupling of $\mu$-GA and SPSS within the WRF-Noah LSM ensemble system

The RF tuning parameters, composed of amplitude and decorrelation length and time scale, are crucial in stochastic perturbation
schemes, but they are commonly determined empirically and have attracted relatively less attention. Generally, the following
three scales are recommended for the tuning parameters: decorrelation length scale of 500, 1000, and 2000 km, decorrelation
time scales of 6 hours, 3 days, and 30 days, and standard deviation of 0.52, 0.18, and 0.06, respectively (see Leutbecher et al.,
2017; Lang et al., 2021). The suitability of these tuning parameters, however, is not certified for SPSS. In this study, we seek
the optimal RF tuning parameters suitable to SPSS using the $\mu$-GA.

Figure 4 describes the coupled system of $\mu$-GA and SPSS within the WRF-Noah LSM ensemble system. The population
size is set to 5 following previous studies (e.g., Lee et al., 2006; Wang et al., 2010). We used 5 ensemble members to represent
the ensemble system because generally 4 to 8 members were expected to be best suited for scientific testing (Leutbecher,





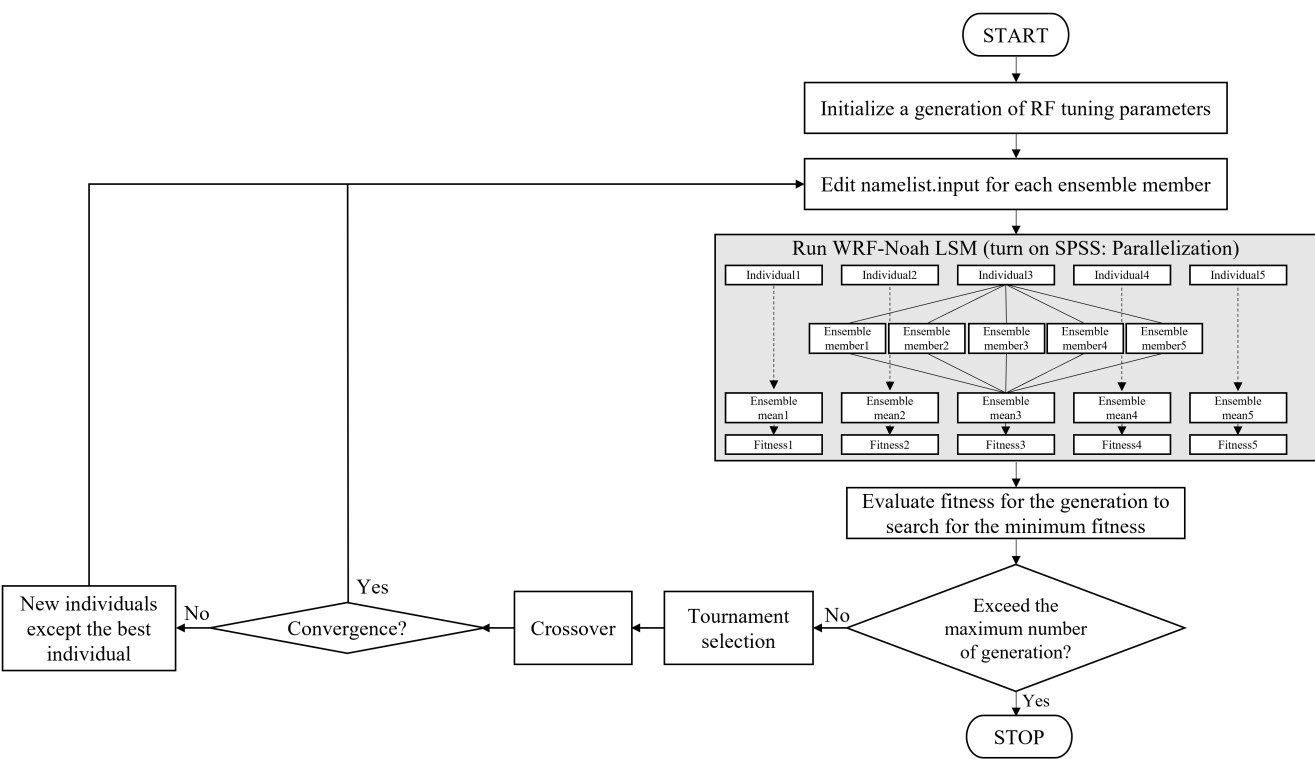

**Figure 4.** Flowchart of the coupled system of $\mu$-GA and SPSS within the WRF-Noah LSM ensemble system.

2019). First, $\mu$-GA randomly initializes RF tuning parameters from the assigned ranges, based on previous studies and our experiments. Next, each individual with 5 ensemble members is configured with different tuning parameters; the model runs for the total population (i.e., 5 individuals) are executed in parallel. We designed a parallel system for individuals (e.g., Lim et al., 2022) to reduce the computational time. When the 6 hour forecasts are completed, the ensemble mean in each population
is made and then evaluated by the fitness function. Among the individuals with different fitness values, $\mu$-GA selects the best individual (i.e., with the smallest fitness value) via elitism and then passes it to the next generation. A new combination of individuals for the next generation is produced by crossover, re-initialization and elitism. Finally, $\mu$-GA repeats these processes during the assigned generations to find the global minimum.

The fitness function is a performance index to evaluate how potential solutions well fit the reference data. The SPSS expects
to improve the performance by increasing the ensemble spread and reducing the ensemble mean error. An additional RF into the soil states naturally increases the ensemble spread; thus, we only examined the reduced ensemble mean error using the root-mean-square error (RMSE), which is a widely used indicator for evaluating the model performance. Then, we calculated the mean squared error (MSE) because the quadratic form is better suited to finding the minimum. Finally, we designed the fitness function using the normalized MSE to explain the interaction between LSM and PBL on the same scale in terms of
accuracy, as follows:



$$fitness = \frac{MSE(x)}{\sigma_{ref(x)}} + \frac{1}{z}\sum_{k=1}^{z}\frac{MSE(y_k)}{\sigma_{ref(y_k)}} \qquad (8)$$

where $x$ is soil temperature or soil moisture and $y$ is temperature or water vapor mixing ratio, $z$ represents a total of 7 vertical levels between 850 and 1000 hPa (i.e., 850, 875, 900, 925, 950, 975, and 1000 hPa) with $k$ the corresponding vertical index, and $\sigma_{ref}$ is the standard deviation of reference data (e.g., Global Forecast System (GFS) analysis) used for verification.

### 2.4 Ensemble Data Assimilation System

Ensemble Kalman filter (EnKF; Evensen, 1994; Whitaker and Hamill, 2002; Houtekamer and Zhang, 2016) uses an ensemble of forecasts to estimate the BEC in the Kalman filter. Based on the Monte Carlo approach, it produces a set of random samples for the analysis and background state probability distributions (Buehner, 2005). We used EnKF (v1.3) provided in the Gridpoint Statistical Interpolation (GSI) community (v3.7) composed of two parts, GSI observer and EnKF (Liu et al., 2018): the GSI observer computes the observation innovations (i.e., observation − background) using the observation operator, and EnKF generates the analysis of each ensemble member.

The traditional Kalman filter update equation can be written as follows:

$$\mathbf{x}^a = \mathbf{x}^b + \mathbf{K}(\mathbf{y}^o - \mathbf{H}\mathbf{x}^b), \qquad (9)$$

$$\mathbf{K} = \mathbf{P}^a\mathbf{H}^T(\mathbf{H}\mathbf{P}^b\mathbf{H}^T + \mathbf{R})^{-1} \qquad (10)$$

where $\mathbf{x}^a$ is analysis on model grids, $\mathbf{x}^b$ is background on model grids, $\mathbf{y}$ is observations, $\mathbf{K}$ is the Kalman gain matrix, $\mathbf{H}$ is the observation operator which transforms data from model space to observation space, $\mathbf{R}$ is the observation error covariance matrix, and $\mathbf{P}^b$ is a flow-dependent BEC matrix estimated using the sample covariance from the ensemble of forecasts. The GSI/EnKF provides two algorithms (i.e., a serial ensemble square root filter (EnSRF) (Whitaker and Hamill, 2002) and a local ensemble Kalman filter (LETKF) (Hunt et al., 2007)) to calculate the analysis increment. The current implemented algorithm is EnSRF, which avoids sampling errors by perturbing observations (Whitaker and Hamill, 2002). Therefore, above Eq. (9) and Eq. (10) are replaced as follows:

$$\mathbf{x}^{'a} = \mathbf{x}^{'b} - \widetilde{\mathbf{K}}\mathbf{H}\mathbf{x}^{'b} \qquad (11)$$

where

$$\widetilde{\mathbf{K}} = \alpha\mathbf{K}, \qquad (12)$$





$\alpha = (1 + \sqrt{(\dfrac{\mathbf{R}}{\mathbf{HP}^b\mathbf{H}^T + \mathbf{R}})})^{-1}$        (13)

where a deviation from the mean is represented by a prime. $\widetilde{\mathbf{K}}$ is the reduced Kalman gain used to update ensemble deviations from the ensemble mean. Here, $\mathbf{R}$ assumes uncorrelated observation error covariance, and $\mathbf{P}^b$ is calculated by $\mathbf{P}^b\mathbf{H}^T$ and $\mathbf{HP}^b\mathbf{H}^T$ from the ensemble of model forecast as follows:

$$\mathbf{P}^b\mathbf{H}^T = \overline{\mathbf{x}'^b(\mathbf{Hx}'^b)^T} = \frac{1}{n-1}\sum\nolimits_{i=1}^{n}\mathbf{x}_i'^b(\mathbf{Hx}_i'^b)^T, \qquad (14)$$

$$\mathbf{HP}^b\mathbf{H}^T = \overline{\mathbf{Hx}'^b(\mathbf{Hx}'^b)^T} = \frac{1}{n-1}\sum\nolimits_{i=1}^{n}\mathbf{Hx}'^b(\mathbf{Hx}'^b)^T \qquad (15)$$

where $n$ is the ensemble size, and $i$ is the index of each ensemble member.

## 3   Data and Experimental Design

### 3.1   Data

The initial and lateral boundary conditions were from the $1.0° \times 1.0°$ National Centers for Environmental Prediction Final
Analysis (NCEP-FNL), which is widely used in the mesoscale model (National Centers for Environmental Prediction, National Weather Service, NOAA, U.S. Department of Commerce, 2000). Because it is based on the GFS with Noah LSM, the land surface boundary conditions are inherently consistent with WRF-Noah LSM (Lin and Pu, 2020). Similarly, the $0.5°$ GFS analysis (National Centers for Environmental Prediction, National Weather Service, NOAA, U.S. Department of Commerce, 2006) was used for reference data in optimization and verification of soil temperature, soil moisture, and atmospheric variables.
Since the number of soil layers and soil depth in GFS analysis are identical to Noah LSM, the interpolation uncertainties are excluded.

Observations of the GSI/EnKF system used the Prepared Binary Universal Form for the Representation of meteorological data (PrepBUFR), which is the prepared conventional observation data for assimilation into GSI (National Centers for Environmental Prediction, National Weather Service, NOAA, U.S. Department of Commerce, 2008). It included upper-air
sounding data from radiosondes, dropsondes, surface stations, ships, buoys, aircrafts, wind profilers, and satellite-based winds. The satellite radiance was not assimilated in this study.

### 3.2   Experimental Design

We designed the following experiments to optimize RF tuning parameters in a coupled system of $\mu$-GA and SPSS within the WRF-Noah LSM ensemble system: OSTP-D and OSTP-N optimized the RF tuning parameters in soil temperature perturbation





at daytime and nighttime, respectively, and OSMP-D and OSMP-N corresponding ones with soil moisture. Experiments were conducted in August, when soil-atmospheric coupling is strongest (Draper, 2021). We ran the 6 hour forecast to generate the BEC in the EDA system. We assumed the daytime started from 00 UTC (09 KST) 1 August 2018, and the nighttime started from 12 UTC (21 KST) 1 August 2018. As for the optimization configuration in $\mu$-GA, we followed recommended settings (Carroll, 1996; Yu et al., 2013; Yoon et al., 2021), e.g., 5 population size, uniform crossover, and 100 generations. A

coupled system of $\mu$-GA and SPSS found a potential solution (e.g., RF tuning parameters) within the assigned ranges (Table 1) by randomly choosing the candidate value among $2^6$, $2^6$, and $2^4$ cases for amplitude, decorrelation length and time scale, respectively. The ensemble ICs (i.e., five ensemble members describing the ensemble system) were produced by the random control variables (CV) method, implemented in the WRF Data Assimilation system (WRFDA). It generated the ensemble ICs by adding the random noise to analysis in the control variable space (Gao et al., 2018); thus, the general perturbation patterns followed the background error. We used the basic option, CV option 3, composed of the following control variables: stream

function ($\phi$), unbalanced velocity potential ($\chi_u$), unbalanced temperature ($T_u$), pseudo relative humidity ($q$), and unbalanced surface pressure ($P_{s,u}$).

Next, we assessed SPSS using optimized tuning parameters as a BEC inflation method in the WRF-GSI/EnKF system. We used 27 ensemble members, considered to be a best ensemble size in terms of accuracy and computational costs in the WRF-

EDA system (Kunii and Miyoshi, 2012). The control variables were u-component wind, v-component wind, surface pressure, virtual temperature, and specific humidity. To prevent a filter divergence, we included the multiplicative inflation method with a 0.9 inflation parameter inflating the analysis ensemble spread back to the one of background. We investigated whether SPSS for soil temperature and soil moisture can alter the ensemble BECs for temperature and water vapor mixing ratio in PBL and the effectiveness of diurnally-varying tuning parameters in the WRF-GSI/EnKF system. The STP1 and STP2 used the daytime

tuning parameters and the diurnally-varying tuning parameters, respectively, for the soil temperature perturbation, while SMP1 and SMP2 represented the soil moisture counterparts. These were compared to the control experiment, referred to as CTRL, representing the current WRF-GSI/EnKF system. All experiments were cycled from 06 UTC 1 August 2018 to 00 UTC 7 August 2018, and the spin-up period was the first 3 days of the total period.

## 4 Results

### 4.1 Optimized RF Tuning Parameters for Soil States on WRF-Noah LSM

Figure 5 shows the converged fitness function (i.e., the minimum of normalized MSE) with respect to the soil temperature and soil moisture perturbations. Optimization results suggested the RF tuning parameters — amplitude, decorrelation length and time scale — at daytime or nighttime (Table 1). As for the OSTP-D, the optimized tuning parameters were 0.13 K amplitude, 2900 km length scale, and 120 s time scale. At the $31^{th}$ generation, the normalized MSE converged to 0.708, having 0.544

for soil temperature in the topsoil layer and 0.164 for averaged temperature in the PBL. The amplitude simply scaled the perturbations without a blow-up issue during the 6 hour forecast. The decorrelation length scale was similar to the domain size (e.g., 3000 km) because soil temperature was affected by solar radiation and varied following the latitude. Since the

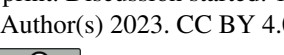



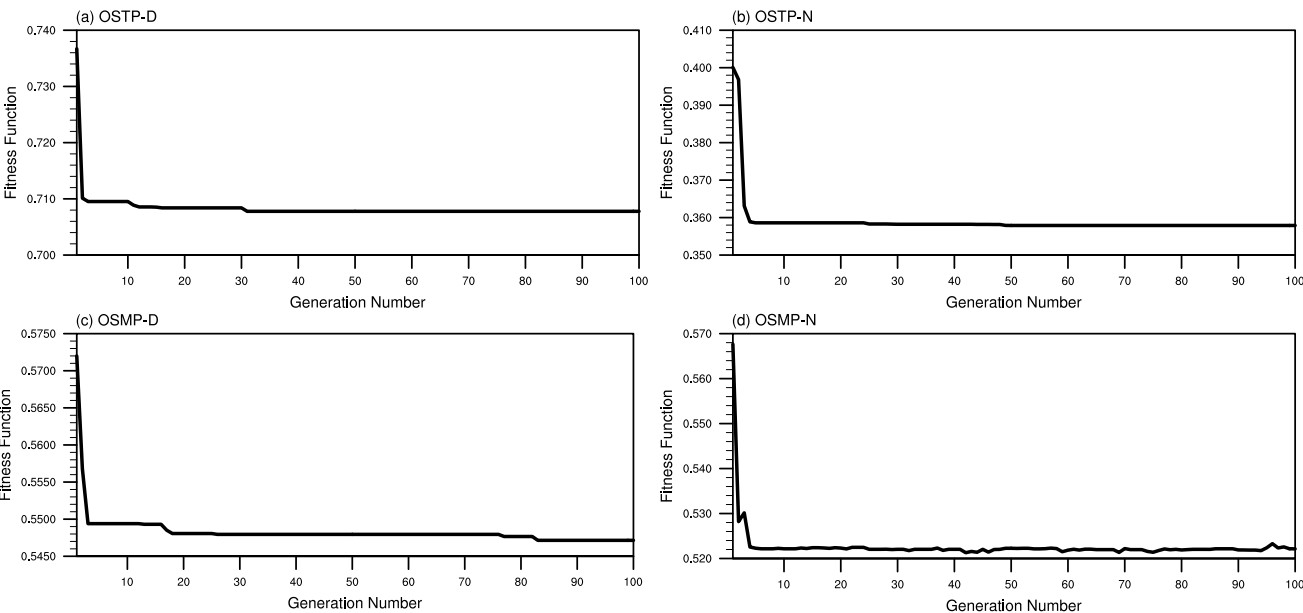

**Figure 5.** The fitness function during assigned maximum generations: (a) OSTP-D, (b) OSTP-N, (c) OSMP-D, and (d) OSMP-N.

decorrelation time scale was inversely proportional to the length scale, the smaller time scale was generated with a larger length scale. As for the OSTP-N, the optimized tuning parameters were 0.01 K amplitude, 100 km length scale, and 900 s

time scale. At the $50^{th}$ generation, the normalized MSE converged to 0.308, having 0.270 for soil temperature in the topsoil layer and 0.088 for the averaged temperature in PBL. The amplitude was much weaker than in the daytime because there was a weaker connection between LSM and PBL at nighttime. The decorrelation length scale was also much smaller than in the daytime, and it was similar to the soil texture distribution involved in the soil temperature calculation (Eq. (1)). On the other hand, the decorrelation time scale was larger than in the daytime due to the smaller length scale at nighttime.

**Table 1.** Optimized RF tuning parameters within the assigned ranges for OSTP-D, OSTP-N, OSMP-D, and OSMP-N.

| RF tuning parameters | Optimization ranges | OSTP-D | OSTP-N | OSMP-D | OSMP-N |
|---|---|---|---|---|---|
| $\sigma$ for soil temperature (in K) | 0.01 - 0.64 | 0.13 | 0.01 | - | - |
| $\sigma$ for soil moisture (in m$^3$ m$^{-3}$) | 0.0001 - 0.0064 | - | - | 0.0003 | 0.0003 |
| $L$ (in km) | 50 - 3200 | 2900 | 100 | 250 | 700 |
| $\tau$ (in s) | 0-900 | 120 | 900 | 900 | 120 |

As for soil moisture, the OSMP-D was optimized at the 0.0003 m$^3$ m$^{-3}$ amplitude, 250 km length scale, and 900 s time scale at daytime and the OSMP-N was optimized at 0.0003 m$^3$ m$^{-3}$ amplitude, 700 km length scale, and 120 s time scale





at nighttime. Because soil moisture was mainly affected by convective rainfall, the decorrelation length scale was optimized at 250 km in the daytime, classified as a mesoscale convection system. The normalized MSE was converged to 0.547 at the $83^{rd}$ generation in the daytime, having 0.362 for soil moisture in the topsoil layer and 0.185 for the averaged water vapor mixing ratio in the PBL. The nighttime produced a larger length scale, 700 km, due to less convection at nighttime. Thus, the decorrelation time scale was also smaller than in the daytime, 120 s. The normalized MSE was converged to 0.521 at the $42^{nd}$ generation, having 0.374 for soil moisture in the topsoil layer and 0.147 for averaged water vapor mixing ratio in the PBL.

### 4.2 BEC inflation using the optimized SPSS within WRF-GSI/EnKF system

Before applying SPSS as a BEC inflation method in the WRF-GSI/EnKF system, we diagnosed the current status of ensembles in CTRL. We examined the zonal mean temperature and water vapor mixing ratio of the ensemble mean error and ensemble BEC: the former is the RMSE of the ensemble mean against 0.5° GFS analysis, and the latter is the ensemble spread composed of a 6 hour forecast (Fig. 6).For temperature, large errors were found over 25-30 °N and near 45 °N, below 925 hPa (Fig. 6a). The ensemble spread was expected to be similar to the ensemble mean error in terms of distribution and magnitude, but it showed underestimated features in the PBL (Fig. 6b). For water vapor mixing ratio, large errors were found in the lower latitude (e.g., 20 - 30 °N) below 750 hPa (Fig. 6c), but the ensemble spread could not express this phenomenon and being larger in 500-750 hPa of 35-45 °N (Fig. 6d). As a result, the overall ensemble spread in CTRL was underestimated compared to the ensemble mean error in PBL. To solve this near-surface uncertainty, we perturbed soil temperature and soil moisture with the optimized RF tuning parameters using SPSS within the WRF-GSI/EnKF system.

Figure 7 shows the time series of ensemble mean error and ensemble spread for soil temperature and temperature. Soil temperature in the topsoil layer showed the diurnal variations in RMSE, but the ensemble spread could not describe the diurnal variations not much and underestimated in CTRL (Fig. 7a). When we perturbed soil temperature, STP1 increased the ensemble spread, but it showed an independent pattern for RMSE, because we only used daytime tuning parameters without the diurnal variations. On the other hand, STP2 described diurnal variations in ensemble spread due to the diurnally-varying tuning parameters. Next, we examined temperature in PBL (e.g., 850 - 1000 hPa) (Fig 7b-d), and it showed the typical problem of underestimated BEC in the EDA system (e.g., filter divergence): the sufficient initial ensemble spread was rapidly reduced as the DA cycle progresses. The STP experiments, including STP1 and STP2, helped to inflate the ensemble spread in temperature, especially at the lowest layer (e.g., 1000 hPa). Since STP1 used a larger single amplitude (e.g., 0.13 K), the ensemble spread was larger than STP2. However, it was hard to find a significant improvement in RMSE in both experiments. Similarly, the surface variables such as 2 m temperature, 2 m water vapor mixing ratio, and 10 m horizontal winds in both experiments showed an increased ensemble spread and almost neutral RMSE (not shown).

Figure 8 shows the time series of ensemble mean error and ensemble spread by the soil moisture perturbations. Unlike the soil temperature behavior (e.g., Fig. 7a), the RMSE of soil moisture was relatively independent of diurnal variations (Fig. 8a). The ensemble spread of soil moisture was smaller than the RMSE at the initial cycle, but it went excessively larger than RMSE as the DA cycles progress. Because we optimized the RF tuning parameters using a 6 hour forecast at the initial cycle (e.g., 06 UTC 1 August 2018) where happened underestimated the ensemble spread, the soil moisture perturbations generated



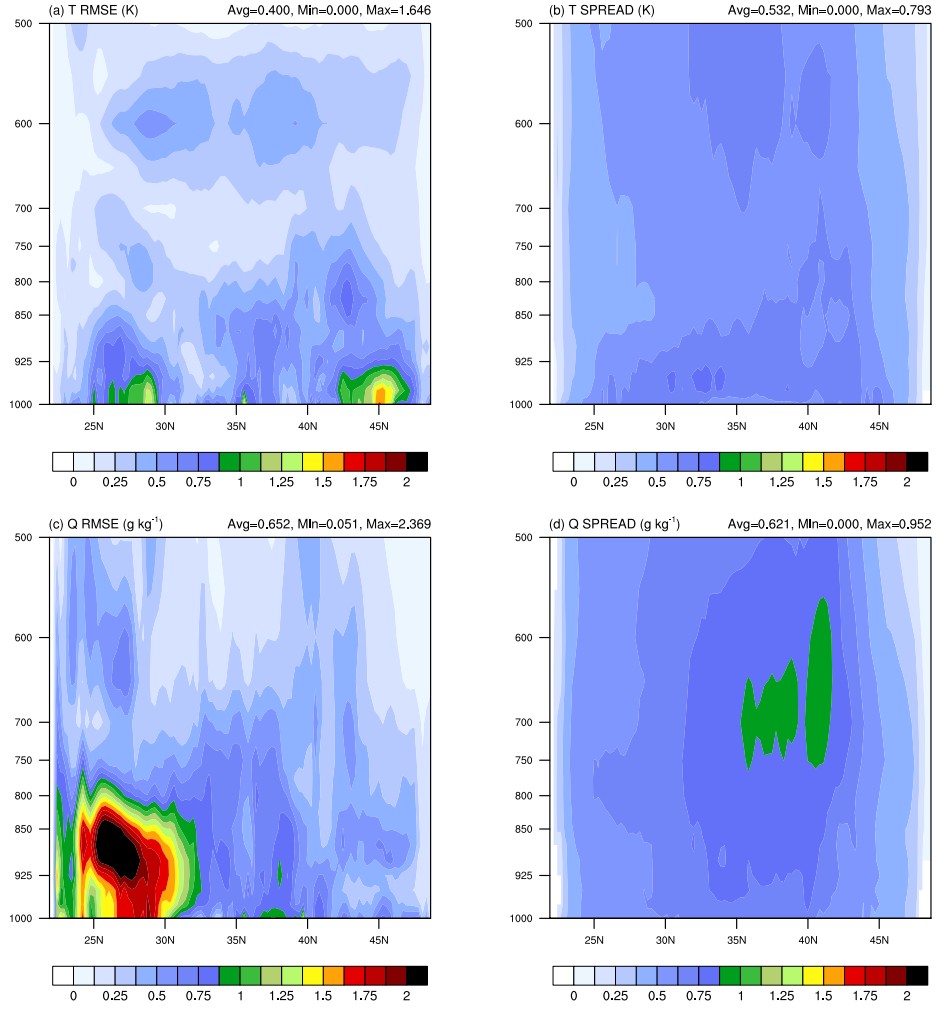

**Figure 6.** The zonal mean ensemble mean error (left panels) and ensemble spread (right panels) for temperature (K; top panels) and water vapor mixing ratio (g kg$^{-1}$; bottom panels) in CTRL over the land during the experimental periods.

excessive ensemble spread during the DA cycles. Consequently, it is recommended optimizing the RF tuning parameters within a cycling system when sufficient computing resources are available. As for water vapor mixing ratio in PBL (Fig. 8b-d), the ensemble spread was also smaller than RMSE, indicating the necessary for BEC inflation in CTRL. Although the amplitude at daytime and nighttime was the same in the soil moisture perturbations (e.g., 0.0003 m$^3$ m$^{-3}$), SMP1 with a smaller length

scale (e.g., 250 km) and longer time scale (e.g., 900 s) made the larger ensemble spread compared to SMP2. Regarding the RMSE changes, the SMP experiments induced an RMSE reduction, especially at 1000 hPa. As for the surface variables such as 2 m temperature, 2 m water vapor mixing ratio, and 10 m horizontal winds, they also showed an increased ensemble spread and weakly decreased RMSE in both experiments (not shown).



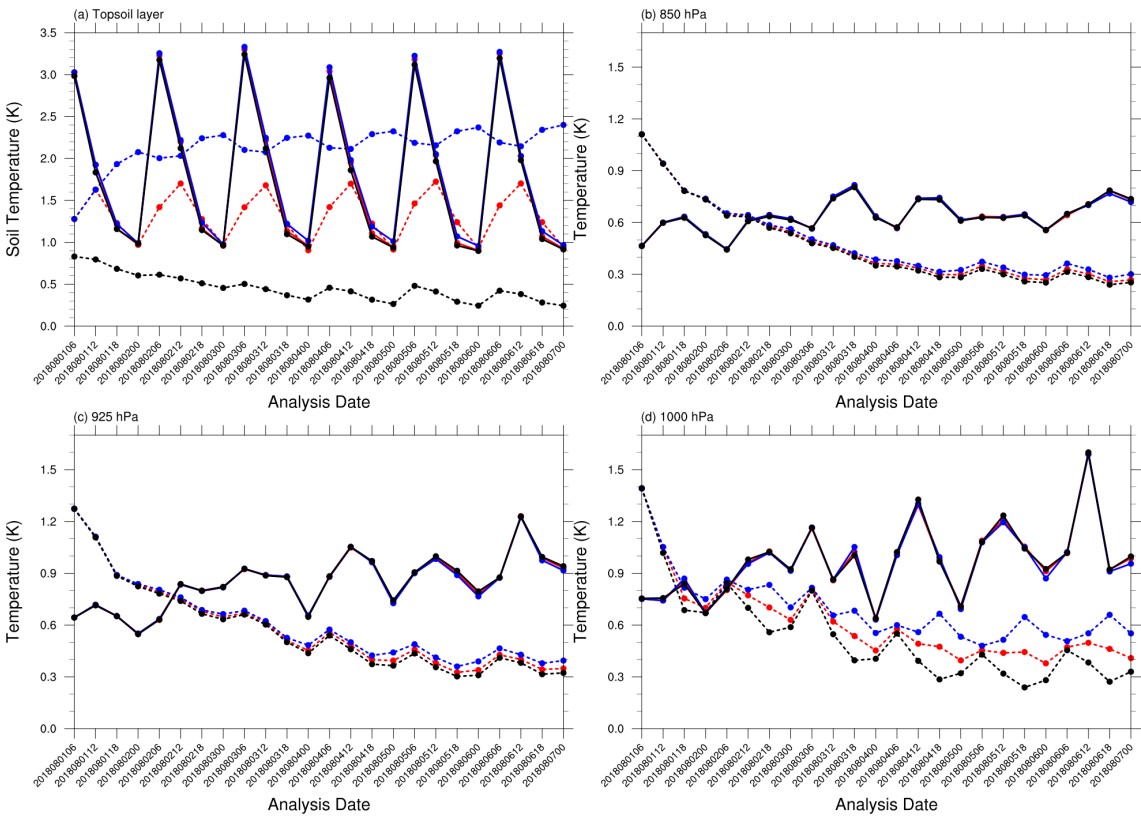

**Figure 7.** Time series of ensemble mean error (solid line) and ensemble spread (dotted line) in CTRL (black), STP1 (blue), and STP2 (red) for background during experimental periods: (a) soil temperature (K) at the topsoil layer, atmospheric temperature (K) over the land at (b) 850 hPa, (c) 925 hPa, and (d) 1000 hPa.

Figure 9 is the scatter plot to assess the relationship between ensemble spread and error: the ensemble in the left (right) domain represents the underestimated (overestimated) ensemble spread, while the ensemble in the diagonal means a good linear relationship between ensemble spread and error. The soil temperature perturbations alleviated the underestimated ensemble spread of soil temperature (Fig. 9a). STP2 satisfied a good relationship, locating on the diagonal, but STP1 generated the overestimated ensemble spread, located in right domain. It means that soil temperature required diurnally-varying tuning parameters. However, when they were propagated to the atmospheric model in PBL (e.g., 1000 hPa), the single (i.e., daytime only) tuning parameters (e.g., STP1) were more effective in increasing the ensemble spread. Because STP1 used a larger amplitude (e.g., 0.13 K) and length scale (e.g., 2900 km), it produced a larger impact on temperature (Fig. 9b). Closer to the surface layer (not shown), the temperature had a larger RMSE and showed an obvious inflation effect from SPSS compared to the above layers. Because it indirectly changed the temperature with the soil temperature perturbing, the responses in atmospheric layers were relatively weak compared to soil. Therefore, we anticipate that the SPSS will be a supplementary



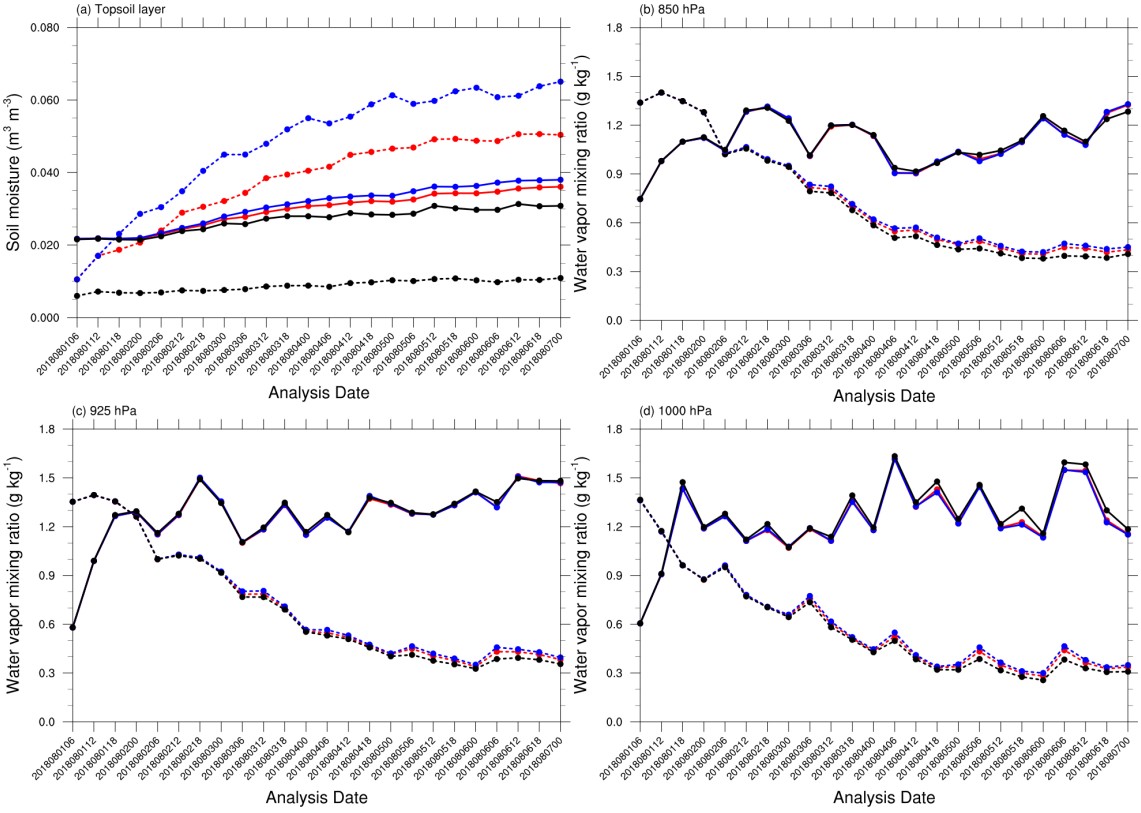

**Figure 8.** Time series of ensemble mean error (solid line) and ensemble spread (dotted line) in CTRL (black), SMP1 (blue), and SMP2 (red) for background during experimental periods: (a) soil moisture (m$^3$ m$^{-3}$) at the topsoil layer, (b) 850 hPa, (c) 925 hPa, and (d) 1000 hPa water vapor mixing ratio (g kg$^{-1}$) over the land.

method to solve the underestimated temperature below the PBL, which was not resolved with the conventional stochastic perturbation scheme (e.g., SPPT).

     As for soil moisture, CTRL was also located in the left domain, showing the underestimated ensemble spread (Fig. 9c). In particular, SMP1 generated apparent overestimation by using the smaller length scale (e.g., 250 km) and larger time scale (e.g., 900 s) under the same amplitude. On the other hand, SMP2 was relatively closer to the diagonal than SMP1, thus the

diurnally-varying tuning parameters were still effective in soil moisture. The propagation to atmospheric states (e.g., water vapor mixing ratio) showed weaker impact compared to the STP experiments (Fig. 9d).

     We expected the BEC inflation to include more observations in the EDA system. We compared the ensemble spread and observation (e.g., rawinsonde) near 1000 hPa, where SPSS showed the greatest impact. For the temperature (Fig. 10a), the ensemble spread in CTRL (black shading) was narrow, so it was hard to contain the observation information. The ensemble

spread, however, was increased in STP1 (blue shading) or STP2 (red shading), so it could include more observations. For the specific humidity (Fig. 10b), the weakly increased ensemble spread in SMP1 (blue shading) or SMP2 (red shading) also tried



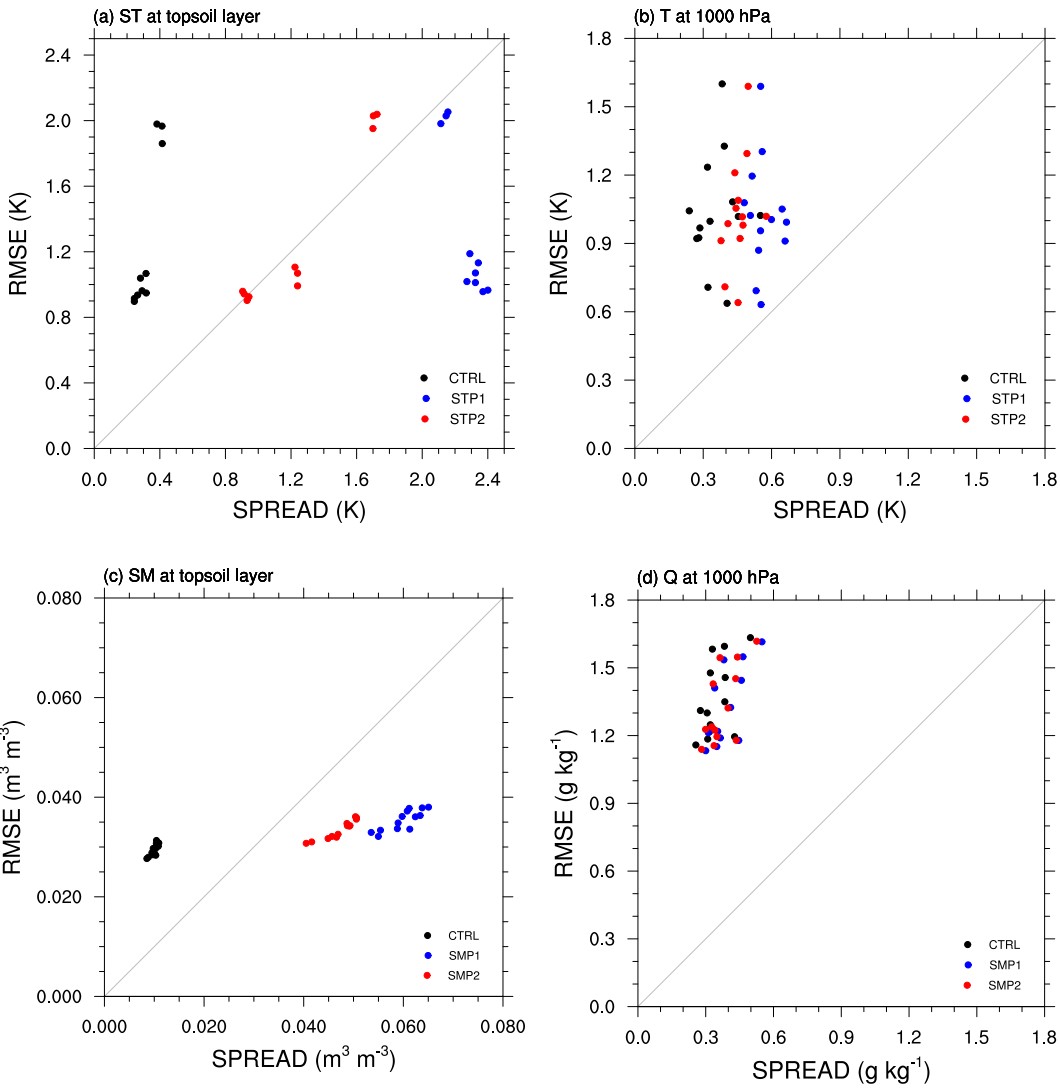

**Figure 9.** Scatter plots of spread and RMSE for background during the experimental periods except for the spin-up period: (a) soil temperature (ST; K) and (b) temperature (T; K) at 1000 hPa from soil temperature perturbations. (c) soil moisture (SM; $m^3$ $m^{-3}$) and (d) water vapor mixing ratio at 1000 hPa (Q; g $kg^{-1}$) from soil moisture perturbations. CTRL is black, STP1 or SMP1 is blue, and STP2 or SMP2 is red. The gray line indicates the diagonal between the ensemble spread (x-axis) and ensemble mean error (y-axis).

to include more observations. Consequently, SPSS may help to increase the number of observations to be assimilated, even if it is negligible.

We also investigated how SPSS affects analysis increment (i.e., analysis – background) that corrects the background using observations and estimates of uncertainties (Buehner, 2005). Figure 11a-c shows the averaged analysis increment and background error of temperature from 850 hPa to 1000 hPa during the experimental period except the spin-up. For the temperature,





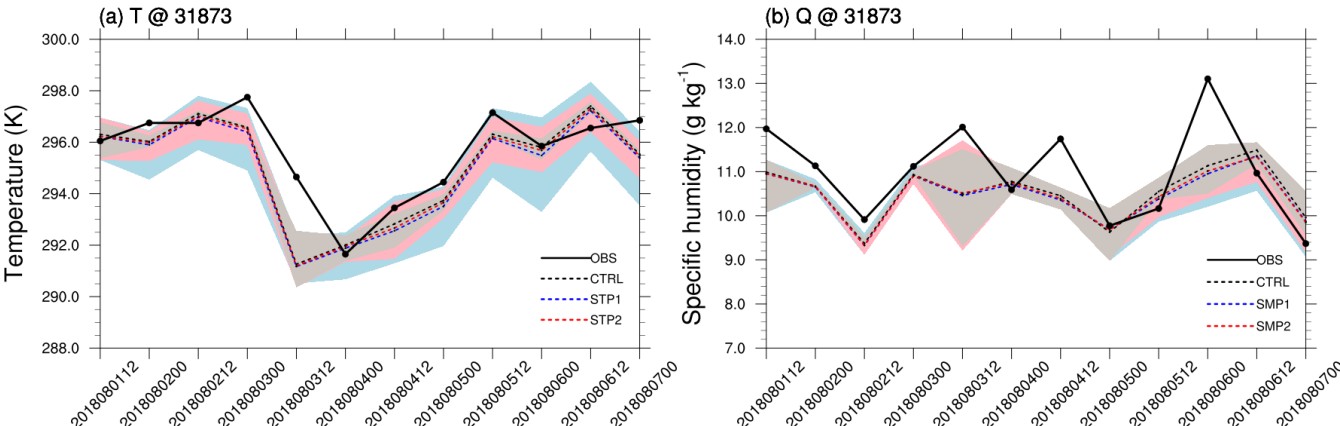

**Figure 10.** Time series of ensemble spread (shading) with the ensemble mean (dotted line) and observation (black solid line) at 31873 station (45.93 °N, 133.74 °E): (a) temperature (K) for CTRL (black), STP1 (blue), and STP2 (red), and (b) specific humidity (g kg$^{-1}$) for CTRL (black), SMP1 (blue), and SMP2 (red).

there was warm bias in north China (NC), northeast China (NE) and most Korean Peninsula (KP) in CTRL, STP1, and STP2, while cold bias in east China (EC), southern region in South Korea (SSK), and most of Japan (JP). To reduce the background error, CTRL produced a negative analysis increment in NC and NE and a positive analysis increment over EC, SSK, and south-

ern Japan (SJP) (Fig. 11a), but they were still insufficient. The STP experiments affected temperature through the sensible and latent heat fluxes and generated an additional analysis increment. For example, STP1 extended the negative increment in NC and NE to reduce the warm bias and the positive analysis increment in EC to reduce the cold bias (Fig. 11b). Moreover, STP1 newly produced negative analysis increments in central Japan (e.g., Kanto) and KP (e.g., Kangwon-do) to reduce the warm bias. STP2 produced a similar analysis increment to STP1, but appeared to have a more localized analysis increment (Fig. 11c).

In the case of soil moisture, most regions had moist bias in all experiments (e.g., CTRL, SMP1, and SMP2) (Fig. 11d-f). In fact, CTRL already had an unreasonable analysis increment increasing humid bias in EC, SSK, and most JP. When soil moisture is perturbing, the humid bias over NC and EC was extended due to broader positive analysis increment (Fig. 11e-f). In both regions, as a result, the soil moisture perturbations increased the background error compared to CTRL. On the other hand, the humid bias over the KP and western Honshu of JP was reduced in both SMP1 and SMP2. In this case, the unreasonable

positive analysis increment over both regions was removed due to soil moisture perturbations. All things considered, the soil moisture perturbations still require additional prescriptions to reduce the background error. For example, we can consider the heterogeneous characteristics such as land-use categories and soil texture in the soil moisture optimization.

Finally, we evaluated the averaged observation innovation statistics of the ensemble prior (i.e., background) to check if the inflation was appropriate in the WRF-GSI/EnKF system. We listed the ensemble spread and the bias and RMSE of innovations

for the control variables (e.g., virtual temperature and specific humidity) (Table 2): When soil temperature was perturbed in STP experiments, the ensemble spread of temperature increased 9.4 % for STP1 (e.g., 0.3189 K) and 3.5 % for STP2 (e.g., 0.3018 K) from CTRL (e.g., 0.2915 K). Accordingly, the warm bias of temperature (0.0598 K) founded in CTRL was reduced





**Figure 11.** The analysis increment (colored contours; positive in red and negative in blue) and the background error against GFS analysis (shaded) for temperature (in K) in (a) CTRL, (b) STP1, and (c) STP2 and for water vapor mixing ratio (in g kg$^{-1}$) in (d) CTRL, (e) SMP1, and (f) SMP2. Results are averaged from 850 hPa to 1000 hPa during the experimental period except the spin-up.

in STP1 (e.g., 0.0532 K) and STP2 (e.g., 0.0555 K) by 11.2 % and 7.2 %, respectively; however the RMSE of innovations was negligible. When soil moisture is perturbed in SMP experiments, the ensemble spread of specific humidity was increased by 2.0 % for SMP1 (e.g., 0.0584 kg kg$^{-1}$) and 1.4 % for SMP2 (e.g., 0.0580 kg kg$^{-1}$) from CTRL (e.g., 0.0573 kg kg$^{-1}$). The dry bias in CTRL (e.g., -0.0066 kg kg$^{-1}$) was reduced in SMP1 (e.g., -0.0059 kg kg$^{-1}$) and SMP2 (e.g., -0.0053 kg kg$^{-1}$) by 10.6 % and 19.5 %, respectively. As for the RMSE of innovations, it was negligible.




**Table 2.** The averaged innovations statistics for ensemble prior including the ensemble spread and bias and RMSE of innovation in the northern hemisphere during the experimental period except the spin-up. The statistics of virtual temperature is from CTRL, STP1, and STP2 and specific humidity is from CTRL, SMP1, and SMP2.

| Control variable | Virtual temperature (K) | | | Specific humidity (kg kg$^{-1}$) | | |
|---|---|---|---|---|---|---|
| Experiment name | CTRL | STP1 | STP2 | CTRL | SMP1 | SMP2 |
| Ensemble spread | 0.2915 | 0.3189 | 0.3018 | 0.0573 | 0.0584 | 0.0580 |
| Bias of innovations | 0.0598 | 0.0532 | 0.0555 | -0.0066 | -0.0059 | -0.0053 |
| RMSE of innovations | 1.4674 | 1.4649 | 1.4661 | 0.1344 | 0.1343 | 0.1342 |

## 5    Conclusions

Soil temperature and moisture perturbations using the stochastic perturbations to soil states scheme (SPSS) can indirectly inflate the ensemble background error covariance (BEC) of temperature and water vapor mixing ratio, respectively, in the planetary boundary layer (PBL) of the regional ensemble data assimilation (EDA) system. To determine the optimal random forcing tuning parameters used in perturbation, we implemented a global optimization algorithm — the micro-genetic algorithm. Each tuning parameter (e.g., amplitude, decorrelation length and time scale) for soil temperature and soil moisture was optimized based on the target time — either daytime or nighttime. First, soil temperature had the amplitude optimized according to the diurnal variation because soil temperature has a diurnal variation greater than soil moisture. Second, the length scale depended on the characteristics of soil variables and target time. During the daytime, the length scale was 2900 km, similar to the domain size, because soil temperature was affected by the solar radiation varying with the latitude. During the nighttime, the length scale was 100 km, which was affected by the soil texture distribution. As for soil moisture, the length scale was 250 km and was classified as a mesoscale convection system due to rainfall effects in the daytime, while the length scale was 700 km due to less convection at nighttime. Lastly, the time scale was inversely proportional to the length scale. As a result, SPSS with diurnal variations in the EDA system depicted a reasonable ensemble spread for soil states, but the propagation to atmospheric states through the sensible and latent heat flux was less effective than the single (i.e., daytime only) tuning parameter. In detail, the soil temperature perturbations in the EDA system reduced the ensemble mean error of temperature in PBL by generating a proper analysis increment during the data assimilation cycles. The soil moisture perturbations, however, strengthened the ensemble mean error of water vapor mixing ratio in PBL by producing an inappropriate analysis increment.

As further studies, we will consider the heterogeneous characteristics of land surface such as land use and soil texture to improve the soil moisture perturbations. Furthermore, we will simultaneously perturb soil temperature and soil moisture in SPSS to improve both temperature and water vapor mixing ratio. Finally, it will be desired to use SPSS in a coupled land-atmosphere data assimilation system where both soil temperature and soil moisture are assimilated. In the coupled data assimilation system, the interaction between land and atmosphere is stronger than in the atmospheric data assimilation system.



Thus, we expect that the improved ensemble spread of soil states can help more to depict the reasonable ensemble BEC in a coupled land-atmosphere data assimilation system.

*Code availability.* The WRF v4.2 source code and run instructions are available at https://github.com/wrf-model/WRF/releases/tag/v4.2 (last access: 10 February 2023) (National Center for Atmospheric Research, 2000b). The WRF Pre-processing system is available at
https://github.com/wrf-model/WPS/releases/tag/v4.2 (last access: 10 February 2023) (National Center for Atmospheric Research, 2000a). The Community GSI v3.7 and EnKF v1.3 is available at https://dtcenter.org/community-code/ensemble-kalman-filter-system-enkf/download (last access: 10 February 2023) (Developmental Testbed Center, 2018). The current version of the GA is available from the website: https://cuaerospace.com/products-services/genetic-algorithm/ga-drive-free-download (last access: 10 February 2023) (Carroll, D. L., 2020). The modified WRF (Section 2.2) is available for download via Zenodo at: https://doi.org/10.5281/zenodo.7622134 (last access: 10 February
2023) (Lim et al., 2023). This repository also includes a copy of WPS v4.2, the exact version of GA driver v1.7a, the GSI/EnKF system v1.3 and the namelists to generate the optimized soil temperature and soil moisture.

*Data availability.* The NCEP-FNL data is available at: https://doi.org/10.5065/D6M043C6 (last access: 10 February 2023) (National Centers for Environmental Prediction, National Weather Service, NOAA, U.S. Department of Commerce, 2000). The GFS analysis used for the verification can be downloaded at: https://www.ncei.noaa.gov/data/global-forecast-system/access/historical/analysis/ (last access: 10 February
2023) (National Centers for Environmental Prediction, National Weather Service, NOAA, U.S. Department of Commerce, 2006). The assimilated PrepBUFR data can also be downloaded at: https://rda.ucar.edu/datasets/ds337.0/ (last access: 10 February 2023) (National Centers for Environmental Prediction, National Weather Service, NOAA, U.S. Department of Commerce, 2008). The all used data (e.g., NCEP-FNL, GFS analysis, and PrepBUFR data) to generate the results in this study is archived at: https://doi.org/10.5281/zenodo.7622134 (last access: 10 February 2023) (Lim et al., 2023).

*Author contributions.* SL, SKP, and CC contributed to conceptualization and designed the experiments. SL developed the model code, contributed to the validation, and prepared the manuscript with contributions from all co-authors. SL and SKP reviewed and edited the manuscript.

*Competing interests.* The authors declare that they have no conflict of interest.

*Acknowledgements.* This research was supported by Basic Science Research Program through the National Research Foundation of Korea
(NRF) funded by the Ministry of Education (2018R1A6A1A08025520) and partly supported by an NRF grant funded by the Korea government (MSIT) (NRF-2021R1A2C1095535). S. Lim was also supported by Basic Science Research Program through the NRF funded by the Ministry of Education (2020R1A6A3A13069223).



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
