# Peer review of "Optimized Stochastic Representation of Soil States Model Uncertainty of WRF (v4.2) in the Ensemble Data Assimilation System"

_Geoscientific Model Development, 2023_

## Referee Comment (RC1)

Review of *"Optimized Stochastic Representation of Soil States Model Uncertainty of WRF (v4.2) in the Ensemble Data Assimilation System"* – Manuscript gmd-2023-28 by S. Lim, S. K. Park, and C. Cassardo

**Summary:**

This study applies an additive stochastic perturbation scheme to soil temperature and moisture (separately) in the topsoil layer of the Noah LSM in the WRF model as a means to improve background error covariances in an ensemble data assimilation system. The micro-genetic algorithm is used to calculate "optimized" amplitude, time, and length scales for different configurations of the perturbation scheme. If the perturbation scheme is working correctly (see MC1), there is some modest indication that it could act as a supplement to other perturbation schemes in the DA cycle (e.g., SPPT, SPP, etc.), since its effect on atmospheric variability is limited when applied in isolation.

There are numerous instances of unclear, incorrect, or otherwise awkward wording that need to be corrected at this stage. Additionally, there are several major questions about the application of both the SPSS and micro-GA that should be addressed before this manuscript can be accepted for publication. Further discussion of the results presented here in the context of other methods for representing land surface uncertainty (see SC2) would be helpful.

**Recommendation:**
Major revisions

**Major Comments:**

1. Figure 3 and Equation 7 do not appear to correspond to the same operation. In Equation 7, the updated state variable is computed as the sum of the random forcing and the original variable. Since the random forcing is small compared to the variables themselves, the differences between panels (7a and 7c) and (7d and 7f) are small, as expected. However, plotted values appear to be larger nearly everywhere in the "Updated" panels (except, for example, near 25N,120E in panel 7f) despite the negative values of the random forcing field. Is this supposed to be the case, if so why? Or, is this a minor figure error (e.g., panels plotted at the wrong time)? Or, is there a larger systematic error in the application of the random forcing scheme?

2. At several points in the manuscript (e.g., Line 373), it is mentioned that the optimized perturbation length scale is related to incoming solar radiation and the spatial scale of soil texture variability, depending on time of day. It might be helpful to provide some support for this (e.g., a map of different land use categories or soil textures as a "Fig. 2b"?). If an experiment were conducted in which the soil texture/land use distribution was artificially made uniform, would the length scale of the nighttime perturbations be more similar to the daytime length scale?

3. Were the optimal perturbation parameters derived over only the two 6-hour periods listed on Lines 226–228? Would using additional periods to optimize these parameters (perhaps averaging over multiple optimization values) improve or degrade performance? Related to this, is it possible to calculate the percent increase in computational time that

optimizing the perturbation parameters adds to the DA process, since re-optimizing the parameters in the cycling system is suggested on Lines 301–302 (of course, this will vary depending on a user's specific model/resource configuration)?

4. There are numerous language and grammar errors that should be revised for clarity and precision. While most of these issues are not severe enough to detract from understanding the research presented in the manuscript, they do interrupt the reader's experience. The manuscript should be thoroughly reviewed for grammar and scientific language, many of these issues are identified in the "technical comments" section.

**Specific Comments:**

1. Line 52: Replace "the less" with "reduced," however, neither choice seems consistent with the discussion of stochastic representations of model uncertainty in this paragraph, since these schemes improve (increase) ensemble spread. Please clarify.

2. Lines 53–58; 61: The sentence starting on line 54 "For example…" could be removed, since the focus of the present study is on soil state perturbations. It would also help to elaborate (in 1–3 sentences) on some of the relevant references included in these lines to show how soil state uncertainty has been represented in recent literature and forecast systems, beyond the already mentioned sensible and latent heat fluxes.

3. Line 85: Please revise the non-scientific text "and so on." Consider changing the in-text list of variables calculated by the Noah LSM to a table.

4. Line 127: The description of the decorrelation time scale is somewhat unclear. As it is written, "determine how long the perturbed errors will be sustained," implies that the perturbations will be held fixed for the duration of the decorrelation time. It may be more clear to say something like, "determine how quickly perturbations evolve in time"

5. Figure 3: Please add a date/time to the caption or figure panels.

6. Line 147–148: Please add a citation in the sentence that begins with "A previous study…"

7. Line 224: First use of OSTP and OSMP acronyms. Please define these in the text.

8. Line 244–246: Please define acronyms when they first appear in the text (STP1, STP2, SMP1, SMP2).

9. Table 1, Lines 161–162, and Line 231:
   a. Please add relevant references (e.g., "previous studies" on line 161) and further discussion to how these ranges were defined.
   b. The numbers of candidate values (Line 231) do not need to be in exponential form. 64, 64, and 16 provides better clarity to the reader.

     c.   Table 1 indicates that a decorrelation time of 0s was included in the tested ranges – would perturbations with this time scale simply behave as temporally uncorrelated noise?

10. Line 274–283 and Figure 6: There is no time and date information in the caption for Figure 6. Are these means and spreads averaged over the entire experimental forecast period (for all 6-hour background forecasts)? Or are they for a single 6 hour forecast? Line 277 refers to "a 6 hour forecast." Please clarify this both in the text and in the figure caption/labels.

11. Line 305; 316; 323–324: There are several occasions where the impact of the tuning parameters on ensemble spread is discussed. On Lines 305 and 323-324, a smaller length scale with a longer timescale increases ensemble spread. On line 316, a larger length scale and larger amplitude scale yield a larger spread. To improve clarity and reduce confusion, it may be better to simply omit the length scale on these lines, since its influence appears to be secondary to the configuration of the time and amplitude scales [which is consistent with SPPT experiments by both Bouttier et al. (2012) and Lupo et al. (2020)].

12. Line 318–319: The sentence starting with "Because…" is unclear and should be revised.
     a.   "…it indirectly changed the temperature…" Does this refer to SPSS indirectly modifying the air temperature? Please clarify.
     b.   "…soil temperature perturbing…" This should probably be "soil temperature perturbation," Please revise.

13. Line 320: Please clarify "underestimated temperature." In the present phrasing, it is unclear if this refers to underestimated temperature spread or a cold bias.

14. Line 327–333: The first and last sentences of this paragraph imply that the number of observations assimilated (or discarded) could be shown here. Elsewhere in the paragraph, the wording seems more related to the ensemble spread including the observed value at station 31873 (Fig. 10). Please consider making a more clear distinction between the uses of "including more observations" in this paragraph. If possible, consider also computing and showing the number of assimilated observations here.

15. Figure 10: Could the shaded areas in Fig. 10 be made partially transparent? This may be helpful at times when the CTRL spread is similar to the STP1 and STP2 spread (e.g., between 20180800212–2018080512 in Fig. 10b.)

16. Line 369–379: It could be worth mentioning here that the amplitude scale of the daytime soil temperature perturbations is an order of magnitude larger than the night time perturbations, which could be a reason why the daytime-only perturbations more effectively propagated to the atmospheric variables. In its current form, the text here focuses more heavily on the length scale and time scale parameters.

17. Line 374: Please rephrase "classified as a mesoscale convection system." The length scale and soil moisture itself is not an MCS, but may be similar in scale. If there are indeed mesoscale convective systems active in the domain at the time of the optimization cycle, it would be helpful to see these on a map, since the optimization appears to be only over a single 6-hour period (e.g., perhaps as a Fig. 2b or 2c).

**Technical comments:**

1. Line 8: Word "respectively" isn't necessary here. This could be a good place in the abstract to specify that soil temperature and soil moisture are perturbed in separate experiments, though, since this is not stated clearly in the abstract.

2. Line 29: Remove unnecessary "The" starting the sentence.

3. Line 32: Replace "out" with "outside"

4. Line 36–38: This sentence would probably be more clear if the colon on line 37 was removed. Either make this into two separate sentences or combine to remove the colon.

5. Line 49: "This…representations" number inconsistency. Please revise.

6. Equations 4 and 5: Are the vertical bars '|' used here to group terms? If so, it is a little confusing when first reading these equations since they look like operators. Consider removing or replacing the vertical bars.

7. Line 113: "resolve the" doesn't seem like the most correct word choice here. "represent" may be more clear.

8. Line 124: "depend on the characteristics being applied" is unclear. Please revise.

9. Lines 129–131: This short paragraph could be merged with the previous one.

10. Line 210: "widely used in the mesoscale model" is somewhat awkward phrasing. Consider revising to "widely used in mesoscale modelling"

11. Line 211–212: "is" "are" number inconsistency, please revise.

12. Line 215: ",…the interpolation uncertainties" – "the" is unnecessary, please remove. Also consider rephrasing to "interpolation uncertainties are avoided"

13. Line 219: "It included…" – Pronoun "It" is inconsistent in number with the antecedent "observations" or "data" in the previous sentence. Consider revision to "These observations included…"

14. Line 221: Unnecessary "The" starting the sentence. Consider revision to "Satellite radiances were…"

15. Line 229: "i.e.," is more appropriate here than "e.g.,". Please revise.

16. Line 230: Parenthetical "e.g.," is probably unnecessary and can be reworded. Consider revision to "a potential solution of RF tuning parameters"

17. Line 277: Missing space after period between "…(Fig. 6).For temperature…"

18. Line 296–308: Numerous instances of awkward or confusing wording, or typos in this paragraph that should be revised.
    a. Line 298: Revise phrasing "it went…"
    b. Line 300: Revise phrasing "when happened underestimated…"
    c. Line 301: Revise phrasing "it is recommended optimizing…"
    d. Line 303: "necessary" should be "necessity"
    e. Line 306–307: Remove "As for the" and ", they"
    f. Line 308: Revise phrasing "weakly decreased RMSE" to "a small RMSE reduction"

19. Line 330: Replace word "tried."

20. Line 337: Add missing words "of the" after most.

21. Line 379: Replace "strengthened" with "increased"

22. Line 381: "As further studies" is awkward phrasing. Consider revising to "in future research"

**References:**
Bouttier, F., B. Vié, O. Nuissier, and L. Raynaud, 2012: Impact of Stochastic Physics in a Convection-Permitting Ensemble. *Mon. Wea. Rev.*, **140,** 3706–3721, https://doi.org/10.1175/MWR-D-12-00031.1.
Lupo, K. M., R. D. Torn, and S. Yang, 2020: Evaluation of Stochastic Perturbed Parameterization Tendencies on Convective-Permitting Ensemble Forecasts of Heavy Rainfall Events in New York and Taiwan. *Wea. Forecasting,* **35,** 5–24, https://doi.org/10.1175/WAF-D-19-0064.1.

---

## Referee Comment (RC2)

Review of GMD-2023-28 titled "Optimized Stochastic Representation of Soil States Model Uncertainty of WRF (v4.2) in the Ensemble Data Assimilation System" by S. Lim, S. K. Park and C. Cassardo

This study used a micro-genetic algorithm to perturb soil moisture and temperature in Noah-LSM within the WRF model (V4.2) to improve ensemble spread that can lead to the short-term prediction in the planetary boundary layer (PBL). Tuning parameters such as the amplitude and the horizontal decorrelation length/time scale of random forcing applied to soil states are evaluated based on 6-h forecasts in temperature and moisture in the boundary layer. While authors claimed that they improved ensemble forecasts by tuning parameters for random perturbations of soil states, this study only worked on an initial ensemble spread, which is different from the actual ensemble spread that grows with cycles. Also, it is not clear how the random perturbations added to soil temperature and moisture can represent the uncertainty of Noah LSM since the actual model uncertainty or error is never examined. The concept of ensemble cycling presented in this draft is either incorrect or vague to make the experiment design and figures not supportive of authors' main points. Numerous fundamental or structural issues are found throughout the manuscript. Unfortunately, authors might need to advance their understanding of the ensemble data assimilation system, as specified in my major comments below. I would recommend authors to spend enough time revisiting the issues, performing ensemble cycling experiments to examine the effect of perturbing land states on the weather prediction, with a clear target time in mind (as six hours is not the characteristic time scale where soil states are expected to significantly affect atmospheric forecasts). For that, I would recommend "Rejection" of the manuscript for now.

Major comments:
1. Lack of innovation: Various stochastic perturbation techniques have been already introduced in the WRF model and widely used to increase ensemble forecast skills (See the references in https://www2.mmm.ucar.edu/wrf/users/docs/user_guide_v4/v4.2/users_guide_chap5.html#stochastic). Authors should recognize all those related efforts specific to the WRF model (including the version 4.2 used in this study) and should justify why we need another perturbation algorithm despite all the corresponding options having been fully supported in the WRF system for a decade, already. For example, users can easily create a 3-D Gaussian random perturbation by simply turning on a namelist parameter (e.g., rand_perturb=1), and the capability of stochastically perturbing parameters also exists for RUC LSM, which can be easily expanded or applicable to Noah LSM. Because these are basically doing the same thing as what authors try to do here, with almost the same tuning parameters, it is mandatory to clarify the need of another algorithm for the same system.

2. Inappropriate references: Along the line, the micro-genetic algorithm should be introduced and understood as an alternative to the existing options available in the WRF model, but in the Introduction section, authors did not include any of the previous work specific to the WRF implementation. It is not convincing if the algorithm introduced in this study or the study *per se* could make any meaningful contributions to improving our understanding or ensemble forecast skills. One should not ignore others' decade-long efforts on the same system for the same problem (e.g., inflating ensemble spread).

3. Inappropriate title: Authors called it optimized representation, but I would expect a much more generic approach to optimize ensemble configurations in the context of a coupled system, not based on a single case study. To me, the presented study rather seems to be one of many ad-hoc tuning practices.

4. Clarity issues: The manuscript needs an extensive work on clarifications. It took me a while to figure out that this study dealt with initial ensemble spread, not the spread during cycling because authors mixed up the two different things throughout the manuscript and from the very beginning. The first statement in the Introduction, for instance, is incorrect: "The ensemble data assimilation

(EDA) describes both initial conditions (ICs) and model uncertainties represented by the flow-dependent background error covariance (BEC)." => In fact, EDA only "requires" initial ensembles to start cycling, and the initial ensembles are not described by EDA because they can be generated separately as this study shows. Also, EDA does not need to describe model uncertainties since there are different ways to construct ensembles without taking model uncertainties into account (e.g., perturbing observations). On the other hand, the general description of EnKF in GSI (e.g., Eqs. (9)-(15)) may not be necessary unless authors made any changes for this application.

5. Improper goal setting with poor experiment design and methodology: From my view, this is one of the most fundamental problems this work has.

   a. The motivation of this study seems to tackle the insufficient ensemble spread that can lead to poor forecast skills. When it comes to under-dispersive ensemble systems, however, the actual problem lies on the reduction of ensemble spread with cycles, not the spread in the initial ensemble. As it takes at least dozens of cycles to saturate the ensemble spread in the regional cycling system, the initial ensemble construction certainly matters to the efficiency (e.g., how quickly the spread grows), but that is a fundamentally different problem from the filter divergence issue where observations are gradually rejected due to the lack of spread.

   b. Moreover, this study focuses on the uncertainties of soil states, it is thus critical to have the land states that are well spun up before the initial time. Considering the imperfect land surface model with no land data assimilation, the characteristic time and spatial scale of soil states, and the initialization from the NCEP-FNL at coarse resolution (1°x1°), a three-day spin-up used in this study is not even close to the very minimum requirement (say, a month).

   c. The experiment design is not well described, but if only 27 ensemble members were used, covariance localization must have been used for such a small ensemble size. In that case, the localization would certainly affect ensemble spread and additive perturbations, but are not found anywhere in this script.

   d. The construction of an initial ensemble needs clarification: Was the random CV option in WRFDA used to perturb atmospheric variables for a 5-member ensemble (L232-235) while the micro-genetic algorithm was used only for soil perturbations in a 27-member ensemble (L239)? How were the two different ensembles combined in your experiment, then? The choice of ensemble size is critical to ensemble spread, but the description of the ensemble system is unclear.

   e. Table 1: How did you decide the optimized ranges for soil moisture and temperature? That range used in this study seems to be ad-hoc, not necessarily representing either model or observation uncertainty.

   f. Authors defined a fitness function in Eq. (8) to determine the best parameter values among several candidates. Although it is true that meteorological variables in the boundary layer are closely tied to land states, atmospheric fields are characterized at different time and spatial scale from that of soil states. Considering the response time of atmospheric variables to soil perturbations as well as many other potential contributors to the boundary layer structure (such as advection, convection, radiation, and clouds), it is questionable if the fitness function based on 6-h forecasts can fully capture the actual impact of soil perturbations on boundary layer forecasts or can be used as a proxy to the optimal parameter settings.

   This is indeed one of the most complex issues to disentangle clearly, but it is my concern that the approach used in this study seems overly simple to resolve such a challenging issue. Anyhow, given that the WRF system already provides various perturbation techniques, another perturbation algorithm cannot guarantee a noble work, and the only way I can see to make this type of work meaningful is to examine how the initial ensemble affects the

ensemble forecast skills for a long period of time in a statistical manner (e.g., not in a single case).

6.  Figures not supportive of main points:
    -   Figure 3: If this study is all about inflating spread, it is expected to show ensemble spread, not a single member of soil states.
    -   Note that an initial ensemble is only a start of cycling and is not supposed to represent the saturated ensemble spread. Hence, Fig. 6 is not needed.
    -   Figures 7, 8, and 10 only show the sensitivity to the initial ensemble, not the optimal ensemble spread.
    -   Figure 9: Again, we do not expect the saturation of ensemble spread at the initial time.
    -   Figure 11: Analysis increments at the initial time are not indicative of the system performance, and the GFS analysis is not quite trustworthy near the surface.

7.  Section 2.1 is named WRF-Noah LSM Coupled System. As far as I know, Noah-LSM is just one of the physics options available in WRF. Did you develop/change anything to enhance the coupling part either in the model or in your analysis step? As all the physics parameterization schemes are interacting with each other within the WRF framework, it is not clear why authors emphasized the "coupled" system here. How does your Noah LSM work differently from all other studies using the same option, again from the modeling or DA aspect?

---

## Author Comment (AC1)

Review of *"Optimized Stochastic Representation of Soil States Model Uncertainty of WRF (v4.2) in the Ensemble Data Assimilation System"* – Manuscript gmd-2023-28 by S. Lim, S. K. Park, and C. Cassardo

**Summary:**

This study applies an additive stochastic perturbation scheme to soil temperature and moisture (separately) in the topsoil layer of the Noah LSM in the WRF model as a means to improve background error covariances in an ensemble data assimilation system. The micro-genetic algorithm is used to calculate "optimized" amplitude, time, and length scales for different configurations of the perturbation scheme. If the perturbation scheme is working correctly (see MC1), there is some modest indication that it could act as a supplement to other perturbation schemes in the DA cycle (e.g., SPPT, SPP, etc.), since its effect on atmospheric variability is limited when applied in isolation.

There are numerous instances of unclear, incorrect, or otherwise awkward wording that need to be corrected at this stage. Additionally, there are several major questions about the application of both the SPSS and micro-GA that should be addressed before this manuscript can be accepted for publication. Further discussion of the results presented here in the context of other methods for representing land surface uncertainty (see SC2) would be helpful.

**Recommendation:**
Major revisions

**Major Comments:**

1. Figure 3 and Equation 7 do not appear to correspond to the same operation. In Equation 7, the updated state variable is computed as the sum of the random forcing and the original variable. Since the random forcing is small compared to the variables themselves, the differences between panels (7a and 7c) and (7d and 7f) are small, as expected. However, plotted values appear to be larger nearly everywhere in the "Updated" panels (except, for example, near 25N,120E in panel 7f) despite the negative values of the random forcing field. Is this supposed to be the case, if so why? Or, is this a minor figure error (e.g., panels plotted at the wrong time)? Or, is there a larger systematic error in the application of the random forcing scheme?

**Response:** Thank you for your comments. In Equation 7, the additive perturbations are added to soil temperature or soil moisture in every time step to forecast. Figure 11 averages the ensemble mean from 00 UTC on 4 August 2018 to 06 UTC on 7 August 2018 using a composite of the 14 cycles' background fields, whereas Figure 3 represents the background field of the first ensemble member at 00 UTC 6 on August 2018. As a result, an individual ensemble member can be exaggerated, as shown in Figure 3; yet, the ensemble mean, which averages 27 ensembles, finally exhibits a moderate perturbation effect. To prevent confusion, we clarified Figure 3 and Figure 11's captions as below:

"Figure 3. The 6 hour forecast field of the first ensemble member, showing the effect of RF on soil temperature (ST in K; upper panels) and soil moisture (SM in $m^3 m^{-3}$; lower panels) at 06 UTC on 1 August 2021"

"Figure 11. The averaged analysis increment (colored contours; positive in red and negative in blue) and the background error against GFS analysis (shaded) for temperature (in K) in (a) CTRL, (b) STP1, and (c) STP2 and for water vapor mixing ratio (in g kg$^{-1}$) in (d) CTRL, (e) SMP1, and (f) SMP2. Results are averaged from 00 UTC on 4 August 2018 to 06 UTC on 7 August 2018 with a composite of the 14 cycles' background fields except the spin-up period and are calculated within vertical layers of 850 hPa to 1000 hPa."

2. At several points in the manuscript (e.g., Line 373), it is mentioned that the optimized perturbation length scale is related to incoming solar radiation and the spatial scale of soil texture variability, depending on time of day. It might be helpful to provide some support for this (e.g., a map of different land use categories or soil textures as a "Fig. 2b"?). If an experiment were conducted in which the soil texture/land use distribution was artificially made uniform, would the length scale of the nighttime perturbations be more similar to the daytime length scale?

**Response:** Thank you for your suggestion. The land use category and soil texture are shown in Figure 2 as subfigures, such as Figure 2(b)-(c), with the following description in Line 81:

"We used the Modified IGBP Moderate Resolution Imaging Spectroradiometer (MODIS) Noah dataset for the land use/land cover categories, and the USDA State Soil Geographic database (STATSGO) for soil texture. The dominant land use category and soil texture are represented in the Figure 2(b)-(c)."

[Figure]

Figure 2. (a) WPS domain configuration, (b) land use category: 1-evergreen needleleaf forest, 2-evergreen broadleaf forest, 3-deciduous needleleaf forest, 4-deciduous broadleaf forest, 5-mixed forests, 6-closed shrublands, 7-open shrublands, 8-woody savannas, 9-savannas, 10-grasslands, 11-permanent wetlands, 12-croplands, 13-urban and built-up, 14-cropland/natural vegetation mosaic, 15-snow and ice, 16-barren or sparsely vegetated, and 17-water, and (c) dominant soil texture: 1-sand, 2-loamy sand, 3-sandy loam, 4-silt loam, 5-silt, 6-loam, 7-sandy clay loam, 8-silt clay loam, 9-clay loam, 10-sandy clay, 11-silty clay, 12-clay, 13-organic material, and 14-water.

As for the artificially uniform soil texture and land use distribution, we can't be certain without experiments. In this manuscript, we simply compared how the optimized length scale was related to the other length scale. Because the optimization experiment using

uniform soil texture and land use categories takes a long time to test, it can be done as a future study to investigate the effect of the length scale on soil temperature and soil moisture perturbation.

3. Were the optimal perturbation parameters derived over only the two 6-hour periods listed on Lines 226–228? Would using additional periods to optimize these parameters (perhaps averaging over multiple optimization values) improve or degrade performance? Related to this, is it possible to calculate the percent increase in computational time that optimizing the perturbation parameters adds to the DA process, since re-optimizing the parameters in the cycling system is suggested on Lines 301–302 (of course, this will vary depending on a user's specific model/resource configuration)?

**Response:** Yes, we selected the representative for daytime and nighttime. Of course, the optimization during the DA cycles can contain the exact error information during the whole experiment period; however, it takes longer under our configuration. Below is the expected computational time of optimization in the DA process:

In our configuration for the optimization,
- Optimization with a 6 h forecast and 27 ensembles requires about 30 min in 1 generation (5 cpu/1 ensemble)
- Each generation repeats up to 100 generations: 30 min x 100 generations = 3,000 min
- Four experiments (OSTP-D, OSTP-N, OSMP-D, and OSMP-N): 3000 min x 4 = 12,000 min (200 h = 8.3 days)
- We can do parallel runs in two experiments: 12,000/2 min = 6,000 min = 4.17 days

In our configuration for verification in the DA cycles,
- 1 cycle with a 6 h forecast and 27 ensembles requires about 30 min. (4 cpu/ 1 ensemble)
- Total 25 cycles: 30 min * 25 cycles = 750 min (12.5 h = 0.5 days)
- Four experiments (STP1, STP2, SMP1, and SMP2) = 750 min x 4 = 3,000 min (50 h= 2.08 days)

In summary, if we use the DA cycling system to optimize the RF tuning parameters, we require about 104.2 days for optimization. Moreover, the verification for DA cycling systems requires about 2.08 days.
Ex) 750 min x 100 generations x 4 experiments/2 = 104.2 days

4. There are numerous language and grammar errors that should be revised for clarity and precision. While most of these issues are not severe enough to detract from understanding the research presented in the manuscript, they do interrupt the reader's experience. The manuscript should be thoroughly reviewed for grammar and scientific language, many of these issues are identified in the "technical comments" section.

**Response:** We carefully checked the grammar errors according to the "technical comments". Thank you for your detailed opinion.

**Specific Comments:**

1. Line 52: Replace "the less" with "reduced," however, neither choice seems consistent with the discussion of stochastic representations of model uncertainty in this paragraph, since these schemes improve (increase) ensemble spread. Please clarify.

**Response:** Thank you for your comment. We revised Lines 51-52 as below:

"Because it is strongly coupled to the atmospheric state at certain times and in certain places, the reduced land-surface uncertainty may lead to better atmospheric forecasts"

2. Lines 53–58; 61: The sentence starting on line 54 "For example…" could be removed, since the focus of the present study is on soil state perturbations. It would also help to elaborate (in 1–3 sentences) on some of the relevant references included in these lines to show how soil state uncertainty has been represented in recent literature and forecast systems, beyond the already mentioned sensible and latent heat fluxes.

**Response:** Thank you for your comments. The examples of the land parameters in Lines 54–56 have been removed. We included how soil state uncertainty has been represented as follows:

"In particular, the soil states directly affect the near-surface temperature and humidity forecasts through the sensible and latent heat fluxes response (Kim and Hong, 2007; Deng et al., 2016; Lin and Pu, 2020; Sutton et al., 2006; Wang et al., 2010b). If the surface is heated during the daytime, sensible energy transfers to the atmosphere and moisture evaporates from the soil; thus, exact soil states are important within the planetary boundary layer (PBL), influencing convection and precipitation (Sutton et al., 2006)."

**References:**
Sutton, C., Hamill, T. M., and Warner, T. T.: Will perturbing soil moisture improve warm-season ensemble forecasts? A proof of concept. Mon. Weather Rev., 134, 3174–3189, https://doi.org/10.1175/MWR3248.1, 2006.
Wang, Y., Kann, A., Bellus, M., Pailleux, J. and Wittmann, C.: A Strategy for perturbing surface initial conditions in LAMEPS. Atmos. Sci. Lett., 11, 108–113, https://doi.org/10.1002/asl.260, 2010b.

3. Line 85: Please revise the non-scientific text "and so on." Consider changing the in-text list of variables calculated by the Noah LSM to a table.

**Response:** Thank you for your comment. We revised the non-scientific expression in Lines 84-85 as below:

"It calculates land surface variables such as soil temperature, soil moisture, surface skin temperature, canopy water content, snowpack water equivalent content, and snowpack depth in the 4 soil layers"

We added Table 1, which listed the state variables for Noah LSM, and we cited Table 1 in Line 86 as below:

"Among the prognostic variables in Noah LSM (Table 1),…"

Table1. List of state variables in Noah LSM (NCAR, 2020b). Soil temperature, total and unfrozen soil moisture are considered in the multiple soil layers.

| State variables | Unit |
|---|---|
| Canopy moisture content | m |
| Ground/canopy/snowpack effective skin temperature | K |
| Soil temperature | K |
| Total soil moisture content | Volumetric fractions ($m^3$ $m^{-3}$) |
| Unfrozen soil moisture content | Volumetric fractions ($m^3$ $m^{-3}$) |
| Actual snow depth | m |
| Liquid water-equivalent snow depth | m |
| Surface albedo including snow effect | Unitless fraction |
| Surface exchange coefficient for heat and moisture | $m\ s^{-1}$ |
| Surface exchange coefficient for momentum | $m\ s^{-1}$ |

**Reference**: NCAR: WRF Version 4.2 [code], https://github.com/wrf-model/WRF/releases/tag/v4.2, (last access: 23 April 2023), 2020b.

4. Line 127: The description of the decorrelation time scale is somewhat unclear. As it is written, "determine how long the perturbed errors will be sustained," implies that the perturbations will be held fixed for the duration of the decorrelation time. It may be more clear to say something like, "determine how quickly perturbations evolve in time"
**Response:** Thank you for your suggestion. We redefined the length and time scale in Lines 126-127 as follows:

"the horizontal decorrelation length scale (L) determines how errors propagate in an isotropic horizontal direction before the spatial autocorrelation function (ACF) of the perturbation field reaches $e^{-1}$; the decorrelation time scale (τ) determines how quickly perturbations evolve in time before the temporal ACF of the perturbation field reaches $e^{-1}$."

5. Figure 3: Please add a date/time to the caption or figure panels.
**Response:** We added a date and time in Figure 3's caption as below:

"The 6 hour forecast field of the first ensemble member, showing the effect of RF on soil temperature (ST in K; upper panels) and soil moisture (SM in $m^3$ $m^{-3}$; lower panels) at 06 UTC on 1 August 2021: (a) Original ST, (b) RF for ST (with σ = 0.13 K, L = 2900 km, and τ = 120 s), and (c) updated ST (i.e., original + RF); (d) original SM, (b) RF for SM (with σ = 0.0003 $m^3$ $m^{-3}$, L = 250 km, and τ = 900 s), and (f) updated SM."

6. Line 147–148: Please add a citation in the sentence that begins with "A previous study…"
**Response:** We add a citation at the beginning of the sentence as follows:

"Krishnakumar (1990) showed that the small population could sufficiently reach the entire search space by crossover alone."

7. Line 224: First use of OSTP and OSMP acronyms. Please define these in the text.
**Response:** Thank you for pointing this out. We defined the experiments' acronyms.

"(1) Optimization of the RF tuning parameters for Soil Temperature Perturbation at Daytime (OSTP-D); and (2) the corresponding one at Nighttime (OSTP-N); (3) Optimization of the RF tuning parameters for Soil Moisture Perturbation at Daytime (OSMP-D); and (4) the corresponding one at Nighttime (OSMP-N)."

8. Line 244–246: Please define acronyms when they first appear in the text (STP1, STP2, SMP1, SMP2).
**Response** Lines 244-247 have been altered to better clarify the experiments:

"Therefore, we performed the following DA cycling experiments to investigate the effect of SPSS on each soil temperature and soil moisture perturbation: (1) Soil Temperature Perturbation 1 (STP1) perturbs soil temperature using the daytime tuning parameters obtained from OSTP-D, and (2) STP2 perturbs soil temperature using the diurnally-varying tuning parameters obtained from OSTP-D and OSTP-N; (3) Soil Moisture Perturbation 1 (SMP1) perturbs soil moisture using the daytime tuning parameters obtained from OSMP-D, and (4) SMP2 perturbs soil moisture using the diurnally-varying tuning parameters obtained from OSMP-D and OSMP-N; (5) These were compared to the control experiment (CTRL), representing the current WRF-GSI/EnKF system."

9. Table 1, Lines 161–162, and Line 231:
    a. Please add relevant references (e.g., "previous studies" on line 161) and further discussion to how these ranges were defined.
    **Response:** Thank you for your comments. We included the relevant references and include why we choose these ranges in Line 161:

    "First, μ-GA randomly initializes RF tuning parameters from the assigned ranges. We assumed potential tuning parameter ranges for the length scale and standard deviation based on three general scales of tuning parameters (Leutbecher et al., 2017). As for the time scale, however, it was redefined for the SPSS because typical ranges (e.g., 6 hours, 3 days, and 30 days) caused excessive perturbations."

    b. The numbers of candidate values (Line 231) do not need to be in exponential form. 64, 64, and 16 provides better clarity to the reader.
    **Response:** We corrected the exponential form to an integer as follows:

    "A coupled system of μ-GA and SPSS found a potential solution of RF tuning parameters within the assigned ranges (Table 1) by randomly choosing the candidate value among 64, 64, and 16 cases for amplitude, decorrelation length and time scale, respectively."

    c. Table 1 indicates that a decorrelation time of 0s was included in the tested ranges – would perturbations with this time scale simply behave as temporally uncorrelated noise?
    **Response:** Yes, we also considered noise that disappears after the initial perturbation is

generated.

10. Line 274–283 and Figure 6: There is no time and date information in the caption for Figure 6. Are these means and spreads averaged over the entire experimental forecast period (for all 6-hour background forecasts)? Or are they for a single 6 hour forecast? Line 277 refers to "a 6 hour forecast." Please clarify this both in the text and in the figure caption/labels.

**Response:** Thank you for pointing this out. Figure 6 represents the averaged ensemble means and spreads for all background (i.e., 6-hour forecasts) over the entire experimental forecast period. We included a detailed description in the manuscript and caption.

In the manuscript, we add following sentences in Lines 275-277:
"We examined the zonal mean temperature and water vapor mixing ratio of the ensemble mean error and ensemble spread for all background (i.e., 6 hour forecasts) over the entire experimental DA cycling period (Fig. 6): the former is the RMSE of the ensemble mean against 0.5° GFS analysis, and the latter is the ensemble spread."

We revised Figure 6's caption as below:
"The zonal mean ensemble mean error (left panels) and ensemble spread (right panels) for temperature (K; top panels) and water vapor mixing ratio (g kg$^{-1}$; bottom panels) as for the 6 hour forecasts of CTRL over the land. Results are averaged from 06 UTC on 1 August 2018 to 06 UTC on 7 August 2018 with a composite of the 25 cycles' background fields (i.e., 1 cycle per 6 hour)."

11. Line 305; 316; 323–324: There are several occasions where the impact of the tuning parameters on ensemble spread is discussed. On Lines 305 and 323-324, a smaller length scale with a longer timescale increases ensemble spread. On line 316, a larger length scale and larger amplitude scale yield a larger spread. To improve clarity and reduce confusion, it may be better to simply omit the length scale on these lines, since its influence appears to be secondary to the configuration of the time and amplitude scales [which is consistent with SPPT experiments by both Bouttier et al. (2012) and Lupo et al. (2020)].

**Response:** Thank you for your suggestion. We removed the length scale description in Lines 305, 316, and 323-324 as below:

(Lines 304-305) "Although the amplitude of the soil moisture perturbations was the same at daytime and nighttime (e.g., 0.0003 m$^3$ m$^{-3}$), SMP1, with a longer time scale (e.g., 900 s), produced a larger ensemble spread than SMP2."

(Lines 315-316) "Because STP1 used a larger amplitude (e.g., 0.13 K), it produced a larger impact on temperature (Fig. 9b)."

(Lines 323-324) "In particular, SMP1 generated an apparent overestimation by using a larger time scale (e.g., 900 s) under the same amplitude."

12. Line 318–319: The sentence starting with "Because…" is unclear and should be revised.
   a. "…it indirectly changed the temperature…" Does this refer to SPSS indirectly

modifying the air temperature? Please clarify.

**Response:** Yes, so we revised Lines 318-319 as below:

"Because SPSS indirectly modifies the air temperature through the soil temperature perturbation, the responses in the atmospheric layers were relatively weak compared to the soil."

b. "…soil temperature perturbing…" This should probably be "soil temperature perturbation," Please revise.

**Response:** We corrected it (Please see the above response).

13. Line 320: Please clarify "underestimated temperature." In the present phrasing, it is unclear if this refers to underestimated temperature spread or a cold bias.

**Response:** We revised Lines 319-321 to explain the underestimated ensemble spread of temperature as below:

"When a standard stochastic perturbation method (such as SPPT) cannot resolve the underestimated ensemble spread of temperature below the PBL, the SPSS can be used as a supplementary method to inflate the ensemble spread."

14. Line 327–333: The first and last sentences of this paragraph imply that the number of observations assimilated (or discarded) could be shown here. Elsewhere in the paragraph, the wording seems more related to the ensemble spread including the observed value at station 31873 (Fig. 10). Please consider making a more clear distinction between the uses of "including more observations" in this paragraph. If possible, consider also computing and showing the number of assimilated observations here.

**Response:** SPSS contributes to include more observations during the DA cycles; however, the change in number of conventional data from PrepBUFR into the ensemble data assimilation (EDA) system is not significant, as shown in Figure S1, which is the time series of the differences in the number of observations used in DA between SPSS (e.g., STP1 or SMP1) and CTRL. We agree with the reviewer to distinguish between the total number of observations and the single station; thus, we revised the paragraph in Lines 327-333 with a supplementary figure as below:

"We expected the BEC inflation to include more observations in the EDA system; thus, we compared the ensemble spread and observation (e.g., rawinsonde) at station 31873 near 1000 hPa, where SPSS showed the greatest impact. For temperature (Fig. 10a), the ensemble spread in CTRL (black shading) was narrow, so it was hard to contain the observation information. The ensemble spread, however, was increased in STP1 (blue shading) and STP2 (red shading), so the EDA system could include more observations. For specific humidity (Fig. 10b), the ensemble spreads in SMP1 (blue shading) and SMP2 (red shading) have weakly increased --- still including more observations. Most stations showed an increasing trend in the number of observations due to SPSS; it is evident that SPSS can increase the number of observations to be assimilated, though not remarkable (Figure S1)."

[Figure]

Figure S1. Time series of the difference of the number of observations (△nobs) for (a) STP1–
CTRL, (b) STP2-CTRL, (c) SMP1-CTRL, and (d) SMP2-CTRL.

15. Figure 10: Could the shaded areas in Fig. 10 be made partially transparent? This may be
helpful at times when the CTRL spread is similar to the STP1 and STP2 spread (e.g.,
between 20180800212–20180800512 in Fig. 10b.)

**Response**: Thank you for your suggestion. Figure 10 was modified in transparent.

[Figure]

16. Line 369–379: It could be worth mentioning here that the amplitude scale of the daytime
soil temperature perturbations is an order of magnitude larger than the night time
perturbations, which could be a reason why the daytime-only perturbations more
effectively propagated to the atmospheric variables. In its current form, the text here
focuses more heavily on the length scale and time scale parameters.

**Response:** Thank you for your suggestion. We added the following discussion regarding
amplitude scale in Lines 369-379 below (Revised sentences are written in blue font):

"First, soil temperature had the amplitude optimized according to the diurnal variation
because soil temperature has a diurnal variation greater than soil moisture. Since the amplitude
scale of the daytime soil temperature perturbations (e.g., 0.13) is an order of magnitude larger

than the scale of the nighttime perturbations (e.g., 0.01), daytime-only perturbations propagate more effectively to atmospheric variables. Second, the length scale depended on the characteristics of soil variables and target time. During the daytime, the length scale was 2900 km, similar to the domain size, because soil temperature was affected by the solar radiation varying with the latitude. During the nighttime, the length scale was 100 km, which was affected by the soil texture distribution. As for soil moisture, the length scale was 250 km and was classified as a mesoscale convection system due to rainfall effects in the daytime, while the length scale was 700 km due to less convection at nighttime. Lastly, the time scale was inversely proportional to the length scale. As a result, SPSS with diurnal variations in the EDA system depicted a reasonable ensemble spread for soil states, but the propagation to atmospheric states through the sensible and latent heat flux was less effective than the daytime-only tuning parameter. In detail, the soil temperature perturbations in the EDA system reduced the ensemble mean error of temperature in PBL by generating a proper analysis increment during the data assimilation cycles."

17. Line 374: Please rephrase "classified as a mesoscale convection system." The length scale and soil moisture itself is not an MCS, but may be similar in scale. If there are indeed mesoscale convective systems active in the domain at the time of the optimization cycle, it would be helpful to see these on a map, since the optimization appears to be only over a single 6-hour period (e.g., perhaps as a Fig. 2b or 2c).

**Response:** Thank you for your suggestion. Since soil moisture is not exactly an MCS, we revised it as you suggested in Lines 373-375. Indeed, there was MCS at the optimization time (e.g., 06 UTC on 1 August 2018 and 18 UTC on 1 August 2018), we added the supplementary figures showing the satellite image.

"In contrast, soil moisture, revealed different length features; in the daytime, the length scale was 250 km --- similar to the scale of a mesoscale convective system (MCS) --- reflecting the rainfall effect by the MCS (Fig. S2a); in the nighttime, the length scale was 700 km with less effect of the MCS (Fig. S2b). In fact, the MCS was active in the afternoon in southern part of the domain (south-central China) on the date the optimization was performed (see Fig. S2a)."

[Figure]

[Figure]

Figure S2. Satellite image at (a) 06 UTC and (b) 18 UTC on 1 August 2018.

**Technical comments:**

1. Line 8: Word "respectively" isn't necessary here. This could be a good place in the abstract to specify that soil temperature and soil moisture are perturbed in separate experiments, though, since this is not stated clearly in the abstract.
**Response:** We removed "respectively" and clarified the separate experiments that perturb soil temperature and soil moisture, respectively.

"It perturbs soil temperature and soil moisture by adding the random forcing to inflate the ensemble spread. In this study, we investigated the effects of separately perturbed soil temperature and soil moisture in each experiment."

2. Line 29: Remove unnecessary "The" starting the sentence.
**Response:** Thank you for pointing this out. We removed "The" as below:

"Sufficient ensemble sizes can moderately remove the sampling error…"

3. Line 32: Replace "out" with "outside"
**Response:** We corrected it as below:

"The former does not use observations outside of a cutoff distance from a state variable, …"

4. Line 36–38: This sentence would probably be more clear if the colon on line 37 was removed. Either make this into two separate sentences or combine to remove the colon.
**Response:** We removed the colon and made the original sentences into two separate sentences, as follows:

"As one of the inflation methods, the stochastic perturbation scheme represents the model uncertainty by perturbing the tendencies or parameters in the physical parameterization schemes (Buizza et al., 1999; Shutts, 2005; Palmer et al., 2009). It assumes that the NWP model contains inevitable uncertainties in the physical processes by simplifications and approximations."

5. Line 49: "This…representations" number inconsistency. Please revise.
**Response:** We changed it to "These stochastic representations of model uncertainty…"

6. Equations 4 and 5: Are the vertical bars '|' used here to group terms? If so, it is a little confusing when first reading these equations since they look like operators. Consider removing or replacing the vertical bars.
**Response:** We removed the vertical bars in Equations 4 and 5.

7. Line 113: "resolve the" doesn't seem like the most correct word choice here. "represent" may be more clear.
**Response:** We changed "resolved the" to "represent".

8. Line 124: "depend on the characteristics being applied" is unclear. Please revise.
**Response:** We revised Lines 124-125 as below:

"The RF tuning parameters that create a perturbation are defined as follows"

9. Lines 129–131: This short paragraph could be merged with the previous one.
**Response:** We merged the two paragraphs.

10. Line 210: "widely used in the mesoscale model" is somewhat awkward phrasing. Consider revising to "widely used in mesoscale modelling"
**Response:** Thank you for your suggestion. We revised it to "widely used in mesoscale modeling".

11. Line 211–212: "is" "are" number inconsistency, please revise.
**Response:** This sentence indicates the initial and lateral boundary conditions; thus, we changed the original sentence to plural as "Because they are based on the GFS with Noah LSM, …"

12. Line 215: ",…the interpolation uncertainties" – "the" is unnecessary, please remove. Also consider rephrasing to "interpolation uncertainties are avoided"
**Response:** Thank you for your comments. We revised Lines 215-216 as below:

"Since the number of soil layers and soil depth in GFS analysis are identical to Noah LSM, interpolation uncertainties are avoided"

13. Line 219: "It included…" – Pronoun "It" is inconsistent in number with the antecedent "observations" or "data" in the previous sentence. Consider revision to "These observations included…"
**Response:** Thank you for your comments. We revised the pronoun in Line 219 as below:

"These observations included…"

14. Line 221: Unnecessary "The" starting the sentence. Consider revision to "Satellite radiances were…"
**Response:** Thank you for your comments. We revised Line 221 as below:
"Satellite radiances were not assimilated in this study"

15. Line 229: "i.e.," is more appropriate here than "e.g.,". Please revise.
**Response:** We revised it to "i.e.,".

16. Line 230: Parenthetical "e.g.," is probably unnecessary and can be reworded. Consider revision to "a potential solution of RF tuning parameters"
**Response:** Thank you for your comments. We revised Line 230 as below:

"A coupled system of μ-GA and SPSS found a potential solution of RF tuning parameters within the assigned ranges (Table 1)…"

17. Line 277: Missing space after period between "…(Fig. 6).For temperature…"
**Response:** Thank you for pointing this out. We added the space between two sentences.

18. Line 296–308: Numerous instances of awkward or confusing wording, or typos in this paragraph that should be revised.

   a. Line 298: Revise phrasing "it went…"
   **Response:** We revised Lines 298-299 as below:

   "At the beginning of the cycle, the ensemble spread of soil moisture was less than the RMSE, but as the DA cycles progressed, it became excessively greater than RMSE."

   b. Line 300: Revise phrasing "when happened underestimated…"
   **Response:** We revised Lines 299-301 as below:

   "Because we optimized the RF tuning parameters using the underestimated ensemble spread at 06 UTC on 1 August 2018, the excessive inflation was applied during the DA cycles."

   c. Line 301: Revise phrasing "it is recommended optimizing…"
   **Response:** We revised Lines 301-302 as below:

   "Hence, in order to maintain the proper balance between ensemble spread and ensemble error during DA cycles, it will be necessary to optimize the RF tuning parameters within the DA cycling system."

   d. Line 303: "necessary" should be "necessity"
   **Response:** We corrected it.

   e. Line 306–307: Remove "As for the" and ", they"
   **Response:** We revised Lines 306-307 as below:

   "Surface variables such as 2 m temperature, 2 m water vapor mixing ratio, and 10 m horizontal winds also showed…"

   f. Line 308: Revise phrasing "weakly decreased RMSE" to "a small RMSE reduction"
   **Response:** We revised Lines 306-308 as below:

   "Surface variables such as 2 m temperature, 2 m water vapor mixing ratio, and 10 m horizontal winds also showed an increased ensemble spread and a small RMSE reduction in both experiments (not shown)."

19. Line 330: Replace word "tried."
**Response:** We revised Lines 330-332 as below:

"For specific humidity (Fig. 10b), the ensemble spreads in SMP1 (blue shading) and SMP2 (red shading) have weakly increased --- still including more observations."

20. Line 337: Add missing words "of the" after most.
**Response:** We revised Lines 336-338 as below:

"For temperature, there was a warm bias in north China (NC), northeast China (NE), and most of the Korean Peninsula (KP) in CTRL, STP1, and STP2, while there was a cold bias in east China (EC), the southern region of South Korea (SSK), and most of Japan (JP)."

21. Line 379: Replace "strengthened" with "increased"
**Response:** We corrected it in Lines 379-380 as below:

"The soil moisture perturbations, however, increased the ensemble mean error of water vapor mixing ratio in PBL by producing an inappropriate analysis increment."

22. Line 381: "As further studies" is awkward phrasing. Consider revising to "in future research"
**Response:** We corrected it.

**References:**
Bouttier, F., B. Vié, O. Nuissier, and L. Raynaud, 2012: Impact of Stochastic Physics in a Convection-Permitting Ensemble. *Mon. Wea. Rev.*, **140,** 3706–3721, https://doi.org/10.1175/MWR-D-12- 00031.1.
Lupo, K. M., R. D. Torn, and S. Yang, 2020: Evaluation of Stochastic Perturbed Parameterization Tendencies on Convective-Permitting Ensemble Forecasts of Heavy Rainfall Events in New York and Taiwan. *Wea. Forecasting,* **35,** 5–24, https://doi.org/10.1175/WAF-D-19-0064.1.

---

## Author Comment (AC2)

Review of GMD-2023-28 titled "Optimized Stochastic Representation of Soil States Model Uncertainty of WRF (v4.2) in the Ensemble Data Assimilation System" by S. Lim, S. K. Park and C. Cassardo

This study used a micro-genetic algorithm to perturb soil moisture and temperature in Noah-LSM within the WRF model (V4.2) to improve ensemble spread that can lead to the short-term prediction in the planetary boundary layer (PBL). Tuning parameters such as the amplitude and the horizontal decorrelation length/time scale of random forcing applied to soil states are evaluated based on 6-h forecasts in temperature and moisture in the boundary layer. While authors claimed that they improved ensemble forecasts by tuning parameters for random perturbations of soil states, this study only worked on an initial ensemble spread, which is different from the actual ensemble spread that grows with cycles. Also, it is not clear how the random perturbations added to soil temperature and moisture can represent the uncertainty of Noah LSM since the actual model uncertainty or error is never examined. The concept of ensemble cycling presented in this draft is either incorrect or vague to make the experiment design and figures not supportive of authors' main points. Numerous fundamental or structural issues are found throughout the manuscript. Unfortunately, authors might need to advance their understanding of the ensemble data assimilation system, as specified in my major comments below. I would recommend authors to spend enough time revisiting the issues, performing ensemble cycling experiments to examine the effect of perturbing land states on the weather prediction, with a clear target time in mind (as six hours is not the characteristic time scale where soil states are expected to significantly affect atmospheric forecasts). For that, I would recommend "Rejection" of the manuscript for now.

→ We appreciate the valuable and constructive comments, which helped us improve the quality of the manuscript. We carried out three major tasks in this study: i) developing the stochastic perturbations to soil states (e.g., soil temperature and soil moisture) scheme (SPSS); ii) optimizing the random forcing (RF) tuning parameters of SPSS; and iii) applying SPSS to an ensemble data assimilation (EDA) system. As a result, our newly developed SPSS inflates the ensemble spread during the forecast time, not the initial ensemble spread in the EDA system. If our major ideas were not well described, we revised the manuscript, and we respond to your valuable comments below.

Major comments:
1. Lack of innovation: Various stochastic perturbation techniques have been already introduced in the WRF model and widely used to increase ensemble forecast skills (See the references in https://www2.mmm.ucar.edu/wrf/users/docs/user_guide_v4/v4.2/users_guide_chap5.html#stochastic). Authors should recognize all those related efforts specific to the WRF model (including the version 4.2 used in this study) and should justify why we need another perturbation algorithm despite all the corresponding options having been fully supported in the WRF system for a decade, already. For example, users can easily create a 3-D Gaussian random perturbation by simply turning on a namelist parameter (e.g., rand_perturb=1), and the capability of stochasticallyperturbing parameters also exists for RUC LSM, which can be easily expanded or applicable to Noah LSM. Because these are basically doing the same thing as what authors try to do here, with almost the same tuning parameters, it is mandatory to clarify the need of another algorithm for thesame system.

Response: Thank you for your practical and specific comments. As you introduced, the WRF model contains various stochastic perturbations such as stochastically perturbed parameterization tendency scheme (SPPT), stochastic kinetic energy backscatter scheme (SKEB), and stochastically perturbed parameterizations scheme (SPP), as well as stochastic perturbations to the boundary conditions. Ollinaho et al. (2017) suggested that future representations of probabilistic model error should address coupled processes such as the surface

and ocean. In particular, land surface is commonly underestimated in ensemble systems (Lavaysse et al. 2013; Leutbecher et al. 2017; Gehne et al. 2019). Our WRF-GSI/EnKF system also suffered from an underestimated ensemble spread in soil states and near-surface atmospheric variables, so we introduced stochastic perturbation on soil states (e.g., soil moisture and soil temperature). Since soil moisture and soil temperature can directly affect near-surface temperature and humidity forecasts through the heat flux response (Kim and Hong, 2007; Lin and Pu, 2020), previous studies attempted to perturb land surface problems using soil states physics tendencies (Gehne et al., 2019; Draper, 2021), initial soil states (Sutton et al., 2006; Gehne et al., 2019), and direct soil states (Draper, 2021). Although SPP, already implemented in RUC LSM of WRF, also perturbs the land surface parameters (i.e., constant values) to improve the boundary layers, it unfortunately does not perturb variables (i.e., model states). To investigate whether the soil states perturbation can inflate the atmospheric variables in the planetary boundary layer in the WRF-GSI/EnKF system, we developed the stochastic perturbations to soil states scheme (SPSS) in WRF. Note that the RF tuning parameters to determine the perturbation features are still incompletely understood (Lupo et al., 2019), and land and atmosphere have different behaviors in dynamics and error growth (Draper, 2021). Therefore, we propose an optimization strategy to find the optimal RF tuning parameters of soil states to figure out the necessity of special scales of RF tuning parameters for land surface perturbations. Finally, we identified whether SPSS with potentially optimal RF tuning parameters can help inflate the ensemble BEC in the EDA system. To clarify motivation of this work, we revised the paragraph in L49-58 as below:

"These stochastic representations of model uncertainty can address a coupled process (e.g., atmosphere-land surface) where a lack of spread exists in the near-surface variables (Leutbecher et al., 2017). In particular, the land surface model (LSM) interacts with the lower atmosphere as boundary conditions. Because it is strongly coupled to the atmospheric state at certain times and in certain places, the reduced land-surface uncertainty may lead to better atmospheric forecasts (MacLeod et al., 2016). In particular, the soil states directly affect the near-surface temperature and humidity forecasts through the sensible and latent heat flux responses (Kim and Hong, 2007; Deng et al., 2016; Lin and Pu, 2020; Sutton et al., 2006; Wang et al., 2010b). If the surface is heated during the daytime, sensible energy transfers to the atmosphere and moisture evaporates from the soil; thus, exact soil states are important within the planetary boundary layer (PBL), influencing convection and precipitation (Sutton et al., 2006). Previous studies perturbed land surface problems using soil states physics tendencies, initial soil state, direct soil states, and surface parameters to examine the impact on atmospheric ensembles (Sutton et al., 2006; MacLeod et al., 2016; Orth et al., 2016; Gehne et al., 2019; Draper, 2021); however, direct soil state perturbation during the forecasts is not yet implemented in the Weather Research and Forecast (WRF) model."

**Reference:**
Sutton, C., Hamill, T. M., and Warner, T. T.: Will perturbing soil moisture improve warm-season ensemble forecasts? A proof of concept. Mon. Weather Rev., 134, 3174–3189, https://doi.org/10.1175/MWR3248.1, 2006.

Wang, Y., Kann, A., Bellus, M., Pailleux, J. and Wittmann, C.: A Strategy for perturbing surface initial conditions in LAMEPS. Atmos. Sci. Lett., 11, 108–113, https://doi.org/10.1002/asl.260, 2010b.

Gehne, M., Hamill, T. M., Bates, G. T., Pegion, P., and Kolczynski, W.: Land surface parameter and state perturbations in the global ensemble forecast system. Mon. Weather Rev., 147(4), 1319-1340, 2019.

2. Inappropriate references: Along the line, the micro-genetic algorithm should be introduced and understood as an alternative to the existing options available in the WRF model, but in the Introduction section, authors did not include any of the previous work specific to the WRF implementation. It is not convincing if the algorithm introduced in this study or the study *per se* could make any meaningful contributions to improving our understanding or ensemble forecastskills. One should not ignore others' decade-long efforts on the same system for the same problem (e.g., inflating ensemble spread).

**Response:** Thank you for your constructive comments. To supplement the absence of an introduction to the micro-genetic algorithm (μ-GA), we added the following paragraph after L58:

"Parameterizations contain uncertain parameters that can lead to sensitive results; hence, optimal parameter estimation is important to enhance the accuracy of the NWP model. As one of the optimization algorithms, the genetic algorithm (GA) is a global optimization based on the Darwinian principles of natural selection (Holland, 1975; Goldberg, 1989). Standard GA and micro-GA, which is efficient GA with a small population, have been successfully used for parameter optimization of a cumulus parameterization scheme in the fifth-generation Pennsylvania State University (PSU)/National Center for Atmospheric Research (NCAR) Mesoscale Model (MM5) (Lee et al., 2006), a convective parameterization scheme in WRF (Yu et al., 2013), and snow-relaxation parameters in the offline Noah Land Surface Model (Noah LSM) (Lim et al., 2022)."

3. Inappropriate title: Authors called it optimized representation, but I would expect a much more generic approach to optimize ensemble configurations in the context of a coupled system, not based on a single case study. To me, the presented study rather seems to be one of many ad-hoctuning practices.

**Response:** Thank you for your comments. We agree that generalizing optimized tuning parameters is risky because we only used a single case for daytime and nighttime. As a result, we revised the title as below:

"Stochastic Perturbation of Soil Sates Model Uncertainty of WRF (v.2) in the Ensemble Data Assimilation System: Preliminary Design for Optimization of Random Forcing Tuning Parameters"

4. Clarity issues: The manuscript needs an extensive work on clarifications. It took me a while to figure out that this study dealt with initial ensemble spread, not the spread during cycling because authors mixed up the two different things throughout the manuscript and from the very beginning. The first statement in the Introduction, for instance, is incorrect: "The ensemble data assimilation (EDA) describes both initial conditions (ICs) and model uncertainties represented by the flow- dependent background error covariance (BEC)." => In fact, EDA only "requires" initial ensembles to start cycling, and the initial ensembles are not described by EDA because they can be generated separately as this study shows. Also, EDA does not need to describe model uncertainties since there are different ways to construct ensembles without taking model uncertainties into account (e.g., perturbing observations). On the other hand, the general description of EnKF in GSI (e.g., Eqs. (9)-(15)) may not be necessary unless authors made any changes for this application.

**Response:** Thank you for your comments. As step by step, we answered your comments.

i)      This study demonstrated the introduction of SPSS to account for land surface model uncertainty in ensemble forecasts from the WRF-GSI/EnKF system. SPSS inflated the ensemble spread of soil temperature or soil moisture in ensemble forecasts during the DA cycling, not the initial ensemble spread. If we carefully explain our experimental design again, first, optimization experiment finds an optimal RF tuning parameters of the SPSS using a single case on each daytime and nighttime. Second, we turned on the SPSS scheme with optimized tuning parameters in EDA system. As the other stochastic perturbation scheme, SPSS perturbed soil temperature or soil moisture during the forecast time at every time step. In other words, ensembles used to calculate the ensemble BEC further perturb soil temperature or soil moisture to increase near-surface uncertainty. Schematic diagram about EDA system is described as below:

[Figure]

**Figure S1.** Schematic diagram of SPSS application in the WRF-GSI/EnKF system with examples of CTRL and STP1 experiments.

ii)     At the first cycle, the initial ensemble members are generated by the lagged forecast, but the initial conditions (i.e., analysis) in every cycle assimilated by available observation and previous 6-hour forecast information; thus we revised the first sentence as below:

"Ensemble data assimilation (EDA) estimates the flow-dependent background error covariance (BEC) starting from a set of initial conditions (ICs) with available observations (Hamill and Whitaker, 2005)."

**Reference:**

Hamill, T. M., and J. S. Whitaker: Accounting for the error due to unresolved scales in ensemble data assimilation: A comparison of different approaches. Mon. Weather. Rev., 133, 3132–3147, https://doi.org/10.1175/MWR3020.1, 2005.

iii)    Representing model uncertainty in the EDA may affect the estimated ensemble BEC based on ensemble forecasts (Leutbecher et al., 2017).

iv)   As you suggested, we removed the Eqs. (9)-(15) and revised L181-206 as below:

"Ensemble Kalman filter (EnKF; Evensen, 1994; Whitaker and Hamill, 2002; Houtekamer and Zhang, 2016) uses an ensemble of forecasts to estimate the BEC in the Kalman filter. Based on the Monte Carlo approach, it produces a set of random samples for the analysis and background state probability distributions (Buehner, 2005). We used EnKF (v1.3) provided in the Gridpoint Statistical Interpolation (GSI) community (v3.7) composed of two parts, GSI observer and EnKF (Liu et al., 2018): the GSI observer computes the observation innovations (i.e., observation − background) using the observation operator, and EnKF generates the analysis of each ensemble member. The GSI/EnKF provides two algorithms (i.e., a serial ensemble square root filter (EnSRF) (Whitaker and Hamill, 2002) and a local ensemble Kalman filter (LETKF) (Hunt et al., 2007)) to calculate the analysis increment. The current implemented algorithm is EnSRF, which avoids sampling errors by perturbing observations (Whitaker and Hamill, 2002)."

5. Improper goal setting with poor experiment design and methodology: From my view, this is one of the most fundamental problems this work has.
   a. The motivation of this study seems to tackle the insufficient ensemble spread that can lead to poor forecast skills. When it comes to under-dispersive ensemble systems, however, the actual problem lies on the reduction of ensemble spread with cycles, not the spread in the initial ensemble. As it takes at least dozens of cycles to saturate the ensemble spread in the regional cycling system, the initial ensemble construction certainly matters to the efficiency (e.g., how quickly the spread grows), but that is a fundamentally different problem from the filter divergence issue where observations are gradually rejected due to the lack of spread.

   **Response:** Thanks for your comments. We agree that the under-dispersive ensemble spread that causes filter divergence should be addressed by the DA period rather than the initial ensemble. Therefore, we implemented SPSS to add continuously evolving random patterns to soil temperature and soil moisture, allowing the ensemble spread to inflate continuously across consecutive assimilation windows. Some unclear descriptions of the experimental design contributed to misunderstanding our main work, so we modified "3.2. experimental design" as follows.

   "In this study, we carried out two main experiments in this study: i) optimizing the RF tuning parameters of SPSS; and ii) applying SPSS to ensemble forecasts can inflate ensemble BECs during DA cycles. First, we conducted the following optimization experiments in a coupled system of μ-GA and SPSS: (1) Optimization of the RF tuning parameters for Soil Temperature Perturbation at Daytime (OSTP-D); and (2) the corresponding one at Nighttime (OSTP-N); (3) Optimization of the RF tuning parameters for Soil Moisture Perturbation at Daytime (OSMP-D); and (4) the corresponding one at Nighttime (OSMP-N). Experiments were conducted in August, when soil-atmospheric coupling is strongest (Draper, 2021). We ran the 6 hour forecast for the daytime starting at 00 UTC (09 KST) 1 August 2018 and the nighttime starting at 12 UTC (21 KST) 1 August 2018. As for the optimization configuration in μ-GA, we followed recommended settings (Carroll, 1996; Yu et al., 2013; Yoon et al., 2021), i.e., 5 population size, uniform crossover, and 100 generations. A coupled system of μ-GA and SPSS found a potential solution of RF tuning parameters within the assigned ranges

(Table 2) by randomly choosing the candidate value among 64, 64, and 16 cases for amplitude, decorrelation length, and time scale, respectively. The ensemble ICs (i.e., five ensemble members describing the ensemble system) were produced by the random control variables (CV) method, implemented in the WRF Data Assimilation system (WRFDA). It generated the ensemble ICs by adding the random noise to analysis in the control variable space (Gao et al., 2018); thus, the general perturbation patterns followed the background error. We used the basic option, CV option 3, composed of the following control variables: stream function ($\Phi$), unbalanced velocity potential ($\chi_u$), unbalanced temperature ($T_u$), pseudo relative humidity (q), and unbalanced surface pressure ($P_{s,u}$).

Second, we used SPSS with optimized RF tuning parameters in DA cycles to add continually evolving random patterns to soil temperature and soil moisture, allowing the ensemble spread to inflate across consecutive assimilation periods. To prepare the WRF-GSI/EnKF system, we used 27 ensemble members, which were known to be the best ensemble size in terms of accuracy and computational costs (Kunii and Miyoshi, 2012), generated by the random CV option 3. The control variables were u-component wind, v-component wind, surface pressure, virtual temperature, and specific humidity. To prevent filter divergence, we used the multiplicative inflation method with a 0.9 inflation parameter to inflate the analysis ensemble spread back to the background and the covariance localization with a horizontal length scale of 500 km and a vertical length scale of 0.4 scale height, based on distance from the observation. We investigated whether SPSS for soil temperature and soil moisture can alter the ensemble BECs for temperature and water vapor mixing ratio in PBL and the effectiveness of diurnally-varying RF tuning parameters in the DA cycling experiments as follows: (1) Soil Temperature Perturbation 1 (STP1) perturbs soil temperature using the daytime tuning parameters obtained from OSTP-D, and (2) STP2 perturbs soil temperature using the diurnally-varying tuning parameters obtained from OSTP-D and OSTP-N; (3) Soil Moisture Perturbation 1 (SMP1) perturbs soil moisture using the daytime tuning parameters obtained from OSMP-D, and (4) SMP2 perturbs soil moisture using the diurnally-varying tuning parameters obtained from OSMP-D and OSMP-N; (5) These were compared to the control experiment (CTRL), representing the current WRF-GSI/EnKF system. All experiments were cycled from 06 UTC 1 August 2018 to 00 UTC 7 August 2018, and the spin-up period was the first 3 days of the total period."

b. Moreover, this study focuses on the uncertainties of soil states, it is thus critical to have the land states that are well spun up before the initial time. Considering the imperfect land surface model with no land data assimilation, the characteristic time and spatial scale of soil states, and the initialization from the NCEP-FNL at coarse resolution (1°x1°), a three-day spin-up used in this study is not even close to the very minimum requirement (say, a month).

**Response:** Thank you for your comments. As you mentioned, land surface models require sufficient initialization, such as a month. Due to enormous computational time, this preliminary study investigated the impacts of soil temperature and soil moisture perturbations using SPSS for 1 week. In the time series of ensemble spread and ensemble errors for soil temperature and soil moisture (Figures 7-8(a)), the rapidly increased or decreased ensemble errors (solid line) and ensemble spreads (dashed line) were relatively saturated after 3 days, such as from 00 UTC on August 4. Accordingly, we assumed that the first 3 days were the spin-up period. In the future study, we will extend the experimental period by at least 1 month based of this a proof of concept study.

[Figure]

(Left) **Figure 7(a).** Time series of ensemble mean error (solid line) and ensemble spread (dotted line) in CTRL (black), STP1 (blue), and STP2 (red) for background during experimental periods: (a) soil temperature (K) at the topsoil layer over the land.

(Right) **Figure 8(a).** Time series of ensemble mean error (solid line) and ensemble spread (dotted line) in CTRL (black), SMP1 (blue), and SMP2 (red) for background during experimental periods: soil moisture ($m^3\ m^{-3}$) at the topsoil layer over the land.

c.  The experiment design is not well described, but if only 27 ensemble members were used, covariance localization must have been used for such a small ensemble size. In that case, the localization would certainly affect ensemble spread and additive perturbations, but are not found anywhere in this script.

**Response:** Thank you for pointing this out. Since we didn't mention the configuration of covariance localization, we included it in L241-242:

"To prevent filter divergence, we used the multiplicative inflation method with a 0.9 inflation parameter to inflate the analysis ensemble spread back to the background and the covariance localization with a horizontal length scale of 500 km and a vertical length scale of 0.4 scale height, based on distance from the observation."

d.  The construction of an initial ensemble needs clarification: Was the random CV option inWRFDA used to perturb atmospheric variables for a 5-member ensemble (L232-235) while the micro-genetic algorithm was used only for soil perturbations in a 27-member ensemble (L239)? How were the two different ensembles combined in your experiment, then? The choice of ensemble size is critical to ensemble spread, but the description of the ensemble system is unclear.

**Response:** Thank you for pointing this out. The random CV option is used to generate initial ensemble members. The 5 ensembles are used to optimize the RF tuning parameters, while the 27 ensembles are used to describe the EDA system. To clarify the ensemble system, we revised L238-240 as below:

"To prepare the WRF-GSI/EnKF system, we used 27 ensemble members, which were known to be the best ensemble size in terms of accuracy and computational costs (Kunii and Miyoshi, 2012), generated by the random CV option 3."

e.  Table 1: How did you decide the optimized ranges for soil moisture and temperature?

That range used in this study seems to be ad-hoc, not necessarily representing either model or observation uncertainty.

**Response:** Thank you for your comments. We defined an optimization range by widening from the typical three scales of RF tuning parameters for decorrelation length scale and standard deviation (e.g., de-correlation length scales of 500, 1000, and 2000 km; and standard deviations of 0.52, 0.18, and 0.06, respectively) in order to increase the probability that the micro-GA will find the optimal solution. We refined the time scale to smaller values because it is rather sensitive to SPSS. The detailed descriptions are included in L161-162, as below:

"First, μ-GA randomly initializes RF tuning parameters from the assigned ranges. We assumed potential tuning parameter ranges for the length scale and standard deviation based on three general scales of tuning parameters (Leutbecher et al., 2017). As for the time scale, however, it was redefined for the SPSS because typical ranges (e.g., 6 hours, 3 days, and 30 days) caused excessive perturbations."

f.  Authors defined a fitness function in Eq. (8) to determine the best parameter values among several candidates. Although it is true that meteorological variables in the boundary layer are closely tied to land states, atmospheric fields are characterized at different time and spatial scale from that of soil states. Considering the response time of atmospheric variables to soil perturbations as well as many other potential contributors to the boundary layer structure (such as advection, convection, radiation, and clouds), it is questionable if the fitness function based on 6-h forecasts can fully capture the actual impact of soil perturbations on boundary layer forecasts or can be used as a proxy to the optimal parameter settings.

**Response:** Thank you for your valuable comments. First of all, I would like to introduce why we defined our fitness function (Eq. (8)). For example, when soil temperature changes, ground heat flux ($G_0$) is affected by Equation (6). To satisfy the surface energy balance (Eq. (3)), the changed $G_0$ can do repartitioning to sensible (H) and latent heat fluxes. As a result, the changed heat fluxes affect the atmospheric temperature. If we see Eq. (4), at least the potential temperature at the surface ($T_{sfc}$) and the atmospheric temperature at the lowest model level ($T_{air}$) can be adjusted. Finally, the perturbed soil temperature was propagated to the planetary boundary layers even in the 6 hour forecast. Since our interest was in how perturbed soil states can change the ensemble BEC composed of 6 hour forecasts in DA cycles, we only evaluated 6 hour forecasts in our fitness function. In a future study, we will evaluate with a longer lead time to consider the response time of atmospheric variables to soil perturbations as well as many other potential contributors to the boundary layer structure since the user can define a fitness function depending on the objective of optimization; this may improve ensemble forecasts over a longer period.

This is indeed one of the most complex issues to disentangle clearly, but it is my concern thatthe approach used in this study seems overly simple to resolve such a challenging issue. Anyhow, given that the WRF system already provides various perturbation techniques, another perturbation algorithm cannot guarantee a noble work, and the only way I can see tomake this type of work meaningful is to examine how the initial ensemble affects the ensemble forecast skills for a long period of time in a statistical manner (e.g., not in a single case).

**Response:** Thank you for your comments. This study is noteworthy in that it provides a proof-of-concept for a RF tuning parameter optimization applied to the soil state perturbation. Although we only optimized a single case for daytime and nighttime, the RF tuning parameters for soil temperature and soil moisture suggested different values reflecting each physical characteristic. Furthermore, although the RF tuning parameters were ad-hoc, the SPSS was effective in improving ensemble spread in soil states and atmospheric variables in PBL. Based on this study, in a future study, we will include more cases and elaborate a fitness function during optimization to suggest general RF tuning parameters for soil state perturbations.

6. Figures not supportive of main points:
   - Figure 3: If this study is all about inflating spread, it is expected to show ensemble spread, nota single member of soil states.
     **Response:** Thank you for your comments. Figure 3 and its caption were updated from a single ensemble member to an ensemble spread as below:

[Figure]

**Figure 3.** Ensemble spread of soil temperature (ST in K; upper panels) and soil moisture (SM in m$^3$ m$^{-3}$; lower panels) and ensemble mean of RF at 06 UTC on 1 August 2021: (a) original ST, (b) RF applied to ST (with $\sigma$ = 0.13 K, L = 2900 km, and $\tau$ = 120 s), and (c) updated ST (i.e., original + RF); (d) original SM, (b) RF applied to SM (with $\sigma$ = 0.0003 m$^3$ m$^{-3}$, L = 250 km, and $\tau$ = 900 s), and (f) updated SM.

   - Note that an initial ensemble is only a start of cycling and is not supposed to represent the saturated ensemble spread. Hence, Fig. 6 is not needed.
     **Response:** Thank you for your comments; however, this is not an initial ensemble spread inflation study, as we responded to comment #4. Figure 6 is a composite of the 25 cycles' background from 06 UTC on 1 August 2018 to 06 UTC on 7 August 2018. Therefore, Figure 6 is significant to distinguish the under-(or over-)estimated ensemble spread (right panels)

compared to ensemble errors (left panels) in the CTRL experiment. As a result, we can determine which inflation methods are required to inflate where an underestimated ensemble spread was reported. To clarify Figure 6, we revised the figure's caption as below:

"**Figure 6.** The zonal mean ensemble mean error (left panels) and ensemble spread (right panels) for temperature (K; top panels) and water vapor mixing ratio (g kg$^{-1}$; bottom panels) as for the 6 hour forecasts of CTRL over the land. Results are averaged from 06 UTC on 1 August 2018 to 06 UTC on 7 August 2018 with a composite of the 25 cycles' background fields (i.e., 1 cycle per 6 hour)."

- Figures 7, 8, and 10 only show the sensitivity to the initial ensemble, not the optimal ensemble spread.
  **Response:** Thank you for your comments; however, this is also not an initial ensemble spread inflation study, as we responded to comment #4. Figures 7, 8, and 10 show the sensitivity to the SPSS using the optimized RF tuning parameters in ensemble forecasts in every DA cycles. In other words, we optimized the RF tuning parameters, not the ensemble spread. Although the optimized RF tuning parameters are ad-hoc, they were effective in SPSS to perturb soil temperature or soil moisture. In DA cycles, SPSS perturbs soil temperature or soil moisture at every time step for every ensemble member during the 6-hour forecast (i.e., background).

- Figure 9: Again, we do not expect the saturation of ensemble spread at the initial time.
  **Response:** Thank you for your comments, however, we inflated the ensemble spread in ensemble forecasts in every DA cycles as responded to comment #4.

- Figure 11: Analysis increments at the initial time are not indicative of the system performance, and the GFS analysis is not quite trustworthy near the surface.
  **Response:** These are the composites of analysis increments during the DA cycles. Figure 11 shows that SPSS helps to modify the analysis increments to reduce the background errors by inflating the ensemble BECs during the DA cycles. Because the GFS analysis uses the same number of soil layers and depth as Noah LSM, it is able to evaluate on the same grid-point verification without interpolation uncertainties. In future work, we can evaluate our performance with other reanalysis data or in-situ observations.

7. Section 2.1 is named WRF-Noah LSM Coupled System. As far as I know, Noah-LSM is just one of the physics options available in WRF. Did you develop/change anything to enhance the coupling part either in the model or in your analysis step? As all the physics parameterization schemes are interacting with each other within the WRF framework, it is not clear why authors emphasized the "coupled" system here. How does your Noah LSM work differently from all other studies using the same option, again from the modeling or DA aspect?

   **Response:** Thank you for pointing this out. We included our stochastic perturbations scheme (i.e., SPSS) in the Noah LSM code to produce the perturbed soil temperature or soil moisture, but we did not change the other physical parts in Noah LSM. Although we intended to emphasize the interactions between atmospheric and land surfaces, we revised the title of subsection 2.1 and 2.2 (L71 and L112) to prevent misleading the coupled system, as follows:

   "2.1. WRF Configurations"
   "2.2. Stochastic Perturbations to Soil States scheme (SPSS)"